# G-actin provides substrate-specificity to eukaryotic initiation factor 2α holophosphatases

Ruming Chen[1], Cláudia Rato[1†], Yahui Yan[1†], Ana Crespillo-Casado[1], Hanna J Clarke[1], Heather P Harding[1], Stefan J Marciniak[1*], Randy J Read[1*], David Ron[1,2,3*]

[1]Cambridge Institute for Medical Research, University of Cambridge, Cambridge, United Kingdom; [2]Wellcome Trust MRC Institute of Metabolic Science, University of Cambridge, Cambridge, United Kingdom; [3]NIHR Cambridge, Biomedical Research Centre, University of Cambridge, Cambridge, United Kingdom

**Abstract** Dephosphorylation of eukaryotic translation initiation factor 2a (eIF2a) restores protein synthesis at the waning of stress responses and requires a PP1 catalytic subunit and a regulatory subunit, PPP1R15A/GADD34 or PPP1R15B/CReP. Surprisingly, PPP1R15-PP1 binary complexes reconstituted in vitro lacked substrate selectivity. However, selectivity was restored by crude cell lysate or purified G-actin, which joined PPP1R15-PP1 to form a stable ternary complex. In crystal structures of the non-selective PPP1R15B-PP1G complex, the functional core of PPP1R15 made multiple surface contacts with PP1G, but at a distance from the active site, whereas in the substrate-selective ternary complex, actin contributes to one face of a platform encompassing the active site. Computational docking of the N-terminal lobe of eIF2a at this platform placed phosphorylated serine 51 near the active site. Mutagenesis of predicted surface-contacting residues enfeebled dephosphorylation, suggesting that avidity for the substrate plays an important role in imparting specificity on the PPP1R15B-PP1G-actin ternary complex.

*For correspondence: sjm20@cam.ac.uk (SJM); rjr27@cam.ac.uk (RJR); dr360@medschl.cam.ac.uk (DR)

†These authors contributed equally to this work

## Introduction

Reversible phosphorylation of the alpha subunit of translation initiation factor 2 (eIF2a) is pivotal to control of global rates of protein synthesis and to modulating mRNA-specific translation in eukaryotes (*Sonenberg and Hinnebusch, 2009*). Phosphorylated eIF2 inhibits its guanine nucleotide exchange factor, eIF2B, attenuating the translation of most mRNA, whilst the translation of a small subset of mRNAs, with special 5′ untranslated regions, is increased (*Hinnebusch, 2005*). As the latter encode potent transcription factors, such as GCN4 in yeasts and ATF4 in animals, eIF2a phosphorylation activates gene expression programs with broad physiological ramifications: the integrated stress response (ISR) in mammals and its yeast counterpart, the general control response (*Harding et al., 2003*).

Four kinases are known to couple diverse upstream signals to eIF2a phosphorylation (*Ron and Harding, 2007*). PERK restrains protein synthesis in response to unfolded proteins in the endoplasmic reticulum. HRI accomplishes the same in response to heme restriction in developing erythroid precursors, whereas PKR is activated by double-stranded RNA to curtail viral protein synthesis in infected cells. The oldest eIF2a kinase, GCN2, is activated by uncharged tRNAs to restore amino acid balance by ISR activation.

In animal cells, eIF2a phosphorylation is reversed by cellular phosphatase complexes consisting of a protein phosphatase 1 catalytic subunit (PP1) and a substrate-specific regulatory subunit. Two such

**eLife digest** For a cell to build a protein, it must first copy the instructions contained within a gene. A complex molecular machine called a ribosome then reads these instructions and translates them into a protein. This translation process involves a number of steps. Proteins called eukaryotic translation initiation factors (or eIFs for short) coordinate the first step in the process, which is known as 'initiation'.

The eIFs also provide the cell with ways to control how quickly it makes proteins. For example, when a cell is stressed, either by starvation or toxins, it adds a phosphate group onto part of an eIF protein, called eIF2α. This modification makes this eIF protein less able to initiate translation, and so the cell builds fewer proteins and conserves more of its resources during times of stress.

Once the stressful conditions are over, the phosphate group is removed from eIF2α by an enzyme called a phosphatase. This phosphatase contains two subunits: one that recognizes eIF2α and another that removes the phosphate group. However, experiments that attempted to recreate this phosphatase activity using just these two subunits in a test tube failed to generate a working enzyme that specifically targeted the phosphate group of eIF2α. This suggests that in cells this enzyme contains an additional unknown subunit. Now, Chen et al. (and Chambers, Dalton et al.) report the identity of a 'missing' third subunit as a protein known as globular-actin or G-actin.

First, Chen et al. looked at the three-dimensional structure of a two-subunit complex formed from the previously known subunits of the phosphatase enzyme, and confirmed that it could remove phosphate groups from a range of proteins and not just eIF2α. However, when a mixture of other proteins taken from mouse cells was added to this two-subunit complex, the complex could specifically remove the phosphate group on the eIF2α protein.

Further experiments revealed that G-actin was the protein in the mixture that, when added to the two-subunit complex, made it specifically target the eIF2α protein. Chen et al. then used a combination of biochemical and structural biology techniques to investigate the phosphatase activity of the three-subunit complex. These findings suggest a plausible molecular mechanism by which the three-subunit complex becomes selective for its target, but further refinements to the structural work will be needed to critically test these suggestions.

regulatory subunits have been identified in mammals: PPP1R15A (known as GADD34) is encoded by an ISR-inducible gene (*Novoa et al., 2001*; *Ma and Hendershot, 2003*), whereas PPP1R15B (known as CReP) is constitutively present (*Jousse et al., 2003*). Cells lacking PPP1R15A are impaired in recovery of protein synthesis during resolution of the stress response (*Novoa et al., 2003*; *Marciniak et al., 2004*), whereas elimination of PPP1R15B results in developmental impairment and perinatal lethality of mice. Importantly, inactivation of both PPP1R15 isoforms is lethal to cells (*Tsaytler et al., 2011*), but can be rescued by converting serine 51 of eIF2a to the non-phosphorylatable alanine, providing genetic proof for the narrow in vivo substrate specificity of the PPP1R15 family of regulatory subunits (*Harding et al., 2009*).

In mammals, PPP1R15 genes encode proteins of over 600 amino acids, but conspicuous conservation of sequence between the A and B isoform is confined to a stretch of ~70 residues at their C-termini. This is also the only sequence in eIF2a-specific regulatory subunits that is conserved broadly across phyla and is encoded by a separate exon; a conserved feature of the genomic organization of the two isoforms of *PPP1R15*. Viruses have co-opted PPP1R15; presumably to neutralize the effects of host PKR (*He et al., 1996*, *1997*) and conservation amongst these viral proteins is also confined to the same stretch of 70 residues. The preeminence of the PPP1R15 C-terminus is also supported by functional experiments, as over-expression of this portion was the basis of their identification as ISR inhibitors in the first place (*Novoa et al., 2001*; *Jousse et al., 2003*), whereas the extended N-terminal region of PPP1R15 participates in membrane binding, regulating subcellular localization and turnover (*Brush et al., 2003*; *Brush and Shenolikar, 2008*; *Zhou et al., 2011*).

Binding to the PP1 catalytic subunit has been mapped to the C-terminal functional core of the PPP1R15 family (*Connor et al., 2001*; *Novoa et al., 2001*) and is shared by the pared-down viral proteins (*He et al., 1998*). An RVxF motif, common to many PP1 binding proteins, is present in the conserved functional core of the PPP1R15 family members and their viral counterparts, and plays an

important role in assembly of the active holophosphatase complex (*He et al., 1998*; *Connor et al., 2001*; *Novoa et al., 2001*). However, beyond these important insights the basis for substrate-specific dephosphorylation remains largely mysterious.

Here, we report on the in vitro reconstitution of an active eIF2a-directed phosphatase from bacterially-expressed PP1 and PPP1R15 functional core. Our studies reveal the essential role of a third cellular factor, G-actin, in endowing the complex with substrate specificity and provide the first glimpse at the structure of a ternary holophosphatase complex possessing such substrate specificity.

## Results

### The PP1-PPP1R15 binary complex

Alignment of regulatory subunits mediating eIF2a$^P$ dephosphorylation reveals that sequence conservation in the PPP1R15 family is limited to a stretch of ~70 residues in their C-termini (*Figure 1A,B* and *Figure 1—figure supplement 1*), which is also the only conserved region of these proteins predicted to be ordered by DISOPRED (*Ward et al., 2004*) (*Figure 1—figure supplement 2A,B*).

The ability to promote eIF2a$^P$ dephosphorylation can be assayed by the repression of ISR target genes whose induction in stressed cells is dependent on levels of eIF2 phosphorylation. Dual-channel flow cytometry analysis of Chinese Hamster Ovary (CHO) cells stably expressing a *CHOP::GFP* ISR reporter gene (*Novoa et al., 2001*) revealed that introduction of an mCherry fusion of PPP1R15A or PPP1R15B resulted in strong repression of the ISR in cells exposed to tunicamycin, a toxin that promotes protein misfolding in the endoplasmic reticulum to activate the eIF2a kinase PERK (*Harding et al., 1999*). C-terminal fragments limited to the conserved region of PPP1R15 family members [mouse PPP1R15A(539–614) and human PPP1R15B(631–700)] were sufficient for ISR attenuation (*Figure 1C*), confirming that the information necessary to promote eIF2a$^P$ dephosphorylation is conveyed by its conserved C-terminal minimal functional core.

To study the PP1-PPP1R15 binary complex in isolation from other cellular factors, the two proteins were co-expressed in a heterologous *Escherichia coli* system. The C-terminal minimal functional core of PPP1R15 was fused to a cleavable glutathione S-transferase (GST) tag and the co-expressed catalytic PP1 subunit was left untagged. PPP1R15 engaged PP1 in a tight complex, reflected in their co-purification on a glutathione-sepharose affinity resin. Following GST tag removal by TEV proteolysis, the two proteins co-eluted from a size exclusion column at a position predicted of a dimer containing one catalytic and one regulatory subunit (*Figure 2A*).

PPP1R15A and PPP1R15B both formed stable complexes with PP1 (both the PP1A and PP1G isoforms were tested) but crystals suited for structural determination only arose from complexes between human PPP1R15B and mouse PP1G. Furthermore, binary complexes containing the entire C-terminal functional core of PPP1R15 (residues 639–701 in the B isoform and 547–614 in the A isoform) were relatively insoluble (more on this point below) but X-Ray diffracting crystals were obtained for complexes between PP1G 7–300 and human PPP1R15B 631–660 (PDB: 4V0V), PPP1R15B 631–669 (PDB: 4V0W) or PPP1R15B 631–684 (PDB: 4V0X) and their structures were solved by molecular replacement (*Table 1*).

The PPP1R15B peptide could be traced from K$^{639}$, immediately N-terminal of the RVxF motif (K$^{640}$VᴛF in human PPP1R15B), through to W$^{662}$ (*Figure 2B–D*). PPP1R15B followed a trajectory along the surface of PP1G similar to that traced by the regulatory subunits spinophilin (*Ragusa et al., 2010*) and PNUTS/PPP1R10 (*Choy et al., 2014*). As expected, the side chains of V$^{641}$ and F$^{643}$ (of the ʀᴠxꜰ motif) engage hydrophobic crevices on the backside of the catalytic subunit, an interaction common to PP1 regulatory subunits (*Egloff et al., 1997*). From there PPP1R15B winds its way through the C-terminal groove to the front of the catalytic subunit. Along this path PPP1R15B Y$^{650}$ occupies a position similar to spinophilin F$^{459}$ and PNUTS F$^{411}$ whereas PPP1R15B I$^{652}$ occupies a position similar to spinophilin T$^{461}$ and PNUTS F$^{413}$, whose side chains engage the ϕϕ motif binding-site of PP1 (*Choy et al., 2014*). The side chain of PPP1R15B R$^{658}$ engages a deep pocket on the PP1 surface, occupying the same position as spinophilin R$^{469}$ and PNUTS R$^{420}$ and forming a conserved salt bridge with PP1G D$^{71}$ in the 'Arg site', as predicted by Choy and colleagues (*Choy et al., 2014*) (*Figure 2D*). The interactions between PPP1R15B and PP1G were thus confined to residues conserved in all PP1 isoforms, explaining the lack of preference of PPP1R15 for binding different PP1 isoforms.

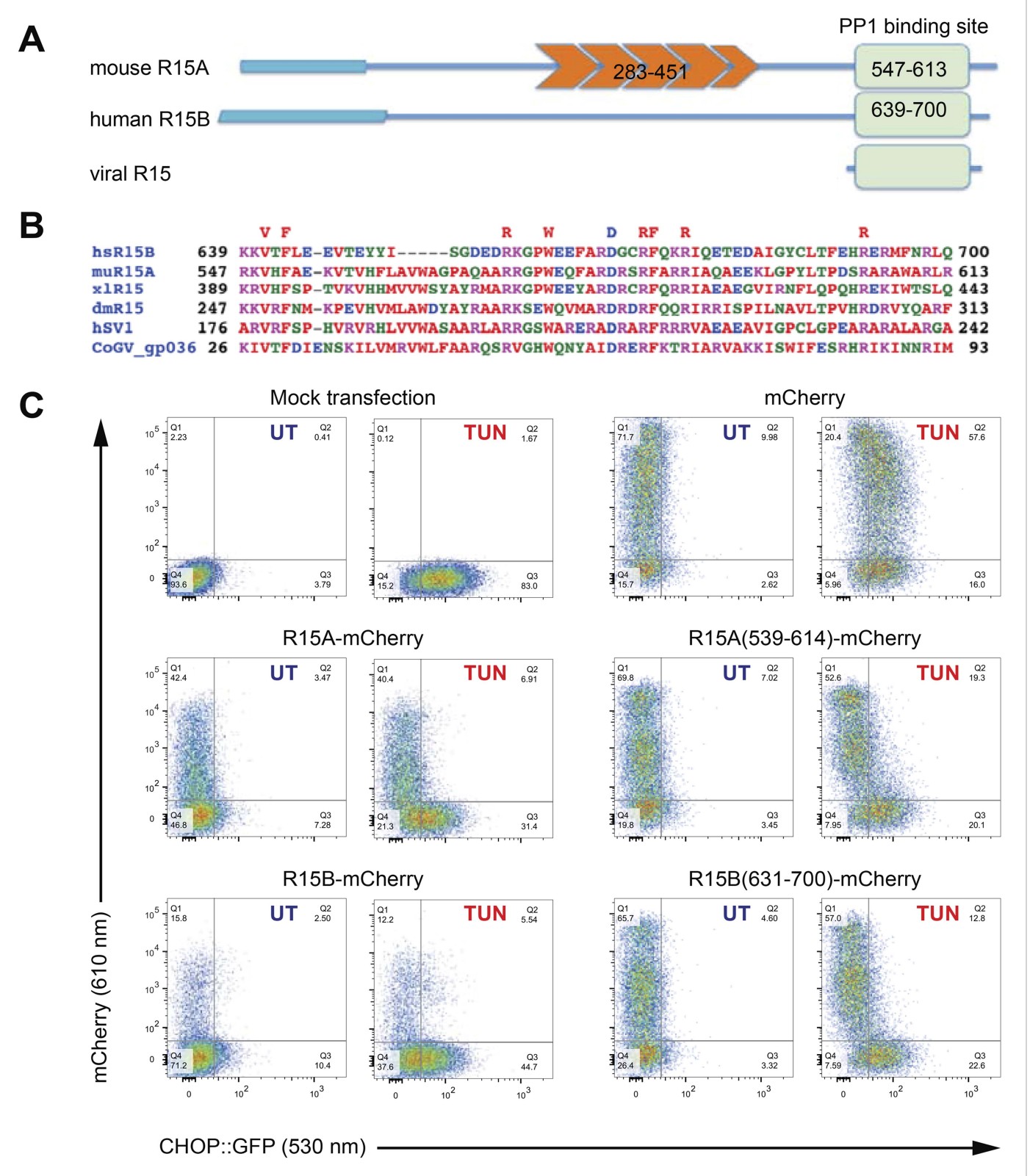

**Figure 1**. A core functional domain in the C-terminus of PPP1R15 family members inhibits the integrated stress response (ISR). (**A**) Cartoon depicting the domain organization of mammalian and viral PPP1R15 proteins. The N-terminal membrane-interaction domain is depicted by a broad line, the repeats, found in the PPP1R15A/GADD34 isoform, are shown in orange and the C-terminal, conserved functional domain that binds the catalytic subunit (PP1) is

*Figure 1. continued on next page*

*Figure 1. Continued*

shown in green. (**B**) Alignment of the conserved 70 residue portion of representative PPP1R15 family members: human PPP1R15B/CReP (hsR15B; UNIPROT: Q5SWA1), mouse PPP1R15A/GADD34 (muR15A; UNIPROT: P17564), frog PPP1R15 (xlR15; UNIPROT: Q9W1E4), fruit-fly PPP1R15 (dmR15; UNIPROT: Q9W1E4), human herpes simplex virus PPP1R15 (HSV1; UNIPROT: P36313), insect virus PPP1R15 (coGV-gp036; UNIPROT: Q1A4R0). Residues conserved in members of the PPP1R15 family across phyla are noted above the alignment. (**C**) Two-dimensional plots of fluorescent intensity of individual CHO cells containing a stably-integrated ISR reporter, *CHOP::GFP* (**Novoa et al., 2001**), following transfection with plasmids encoding mCherry and fusions of PPP1R15 with mCherry. Where indicated the cells were exposed to the ER stress-inducing toxin tunicamycin (TUN). GFP fluorescent intensity, reporting on the activity of the UPR (X-axis) was detected at 530 nm following excitation at 488 nm, whereas mCherry fluorescent intensity, reporting on the level of PPP1R15-mCherry fusion proteins (Y-axis) was detected at 610 nm following excitation at 561 nm. Suppression of the ISR by PPP1R15 is reflected by the accumulation of CHOP:GFP[weak]/mCherry[strong] cells in quadrant 1 (Q1) and is conspicuous in cells transduced with either the full-length PPP1R15 expression vectors (R15A-mCherry and R15B-mCherry) or their C-terminal core functional fragments [R15A(539–614)-mCherry and R15B (631–700)-mCherry].

The following figure supplements are available for figure 1:

**Figure supplement 1**. An extended alignment of PPP1R15 regulatory subunit family members reveals that the conservation of amino acid sequence is confined to a ~70 residue segment.

**Figure supplement 2**. Disordered and ordered regions predicted in PPP1R15 regulatory subunits.

Despite their presence in the crystallized proteins, none of the residues C-terminal to W[662] could be assigned in the density maps, suggesting that the C-terminal portion of PPP1R15's functional core is disordered in the binary complex. Interestingly both spinophilin and PNUTS also appear to disengage from the surface of PP1 at a similar position (D[475] of spinophilin and N[424] of PNUTS), suggesting that the C-terminal extension of these regulatory subunits evolved to engage component(s) missing from the binary complex.

PP1G in these crystals conformed to previous structural determinations of PP1 isoforms (RMS: 0.19–0.29 Å over more than 240 Cα atoms), indicating that the binding of PPP1R15 did not change PP1's structure. Previous structures of PP1 revealed two metal ions, M1 and M2 (likely $Mn^{2+}$) in the shallow groove implicated in catalysis. Two metal ions were also detected in the complex of PP1G 7–300 with PPP1R15B 631–669 (PDB: 4V0W), occupying the sites observed in other crystal forms of PP1 (*Figure 2—figure supplement 1A,B*). However, in the crystal structure of the binary complex of PP1G with the shorter, PPP1R15B 631–660 (PDB: 4V0V), and the longer PPP1R15B 631–684 (PDB: 4V0X), only a single metal ion, occupying an identical M2 position, was observed (*Figure 2—figure supplement 1A*). An unoccupied M1 site was previously observed in the crystal structure of PP1G-Inhibitor-2 complex (PDB: 2O8G) and the okadaic acid-bound PP1G (PDB: 1JK7), suggesting that the metal ion at M1 site is more labile than that at the M2 site and that the structures determined here provide different snapshots of metal ion occupancy of the catalytic subunit.

## Substrate-specificity of the PPP1R15-PP1 complex

The structure of the binary complex of PP1G and PPP1R15B 631–684 (PDB: 4V0X), with ~2800 Å² of buried surface at the interface of its two protomers, is consistent with the high affinity of PPP1R15 for PP1, a property it shares with other regulatory subunits. However, given their similar trajectory on the surface of the catalytic subunit, these structures do not provide a ready answer for the basis of substrate specificity of the different holophosphatases. Therefore, we tried to measure the specificity of binary complexes formed by the complete functional core of PPP1R15 631–701 and PP1. Appending *E. coli* maltose binding protein (MalE) to the functional core of PPP1R15B (631–701) was found to stabilize the binary complex in solution and enabled its recovery in amounts and purity suited for biochemical studies (*Figure 3A*).

Escalating concentrations of binary complex were assayed for their phosphatase activity directed towards two different substrates (presented at similar concentrations): GST[P], consisting of globular glutathione S-transferase with a short unstructured C-terminal peptide derived from the regulatory portion of the transcription factor CREB that is stoichiometrically phosphorylated on a single serine by protein kinase A (*Ron and Dressler, 1992*), served as the reference, non-specific substrate, whereas the soluble N-terminal lobe of human eIF2a (1–185) (*Ito et al., 2004*), phosphorylated on serine 51 by

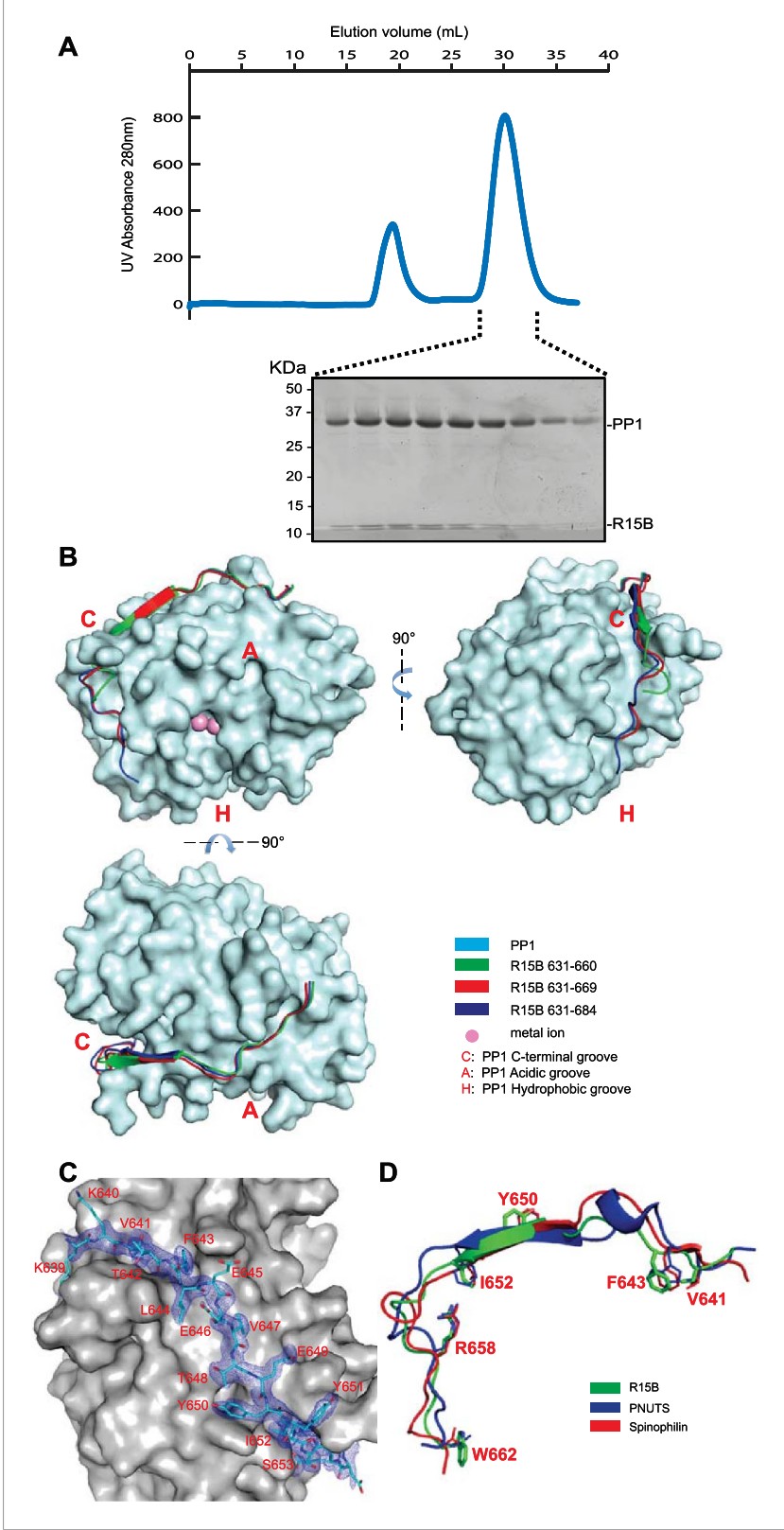

**Figure 2**. PPP1R15 engages the PP1 catalytic subunit through multiple contacts. (**A**) UV protein absorbance trace of a PPP1R15B(630–701)-PP1G(7–300) binary complex expressed in bacteria and resolved by size-exclusion chromatography. The indicated fractions from the chromatogram are presented in the Coomassie-stained
*Figure 2. continued on next page*

*Figure 2. Continued*

SDS-PAGE below. The position of PP1 and the PPP1R15 peptide is indicated. (**B**) Cartoon representation of the indicated segments of human PPP1R15B (in colored ribbon diagram) in complex with mouse PP1G (in cyan surface representation) shown from three perspectives. The position of the hydrophobic (H), C-terminal (C) and acidic (A) groves of PP1 are provided for orientation and the metal ions at the enzymes active site are colored pink. (**C**) Cartoon representation of PPP1R15B residues $K^{639}$ to $R^{653}$ in stick diagram and the corresponding density map, contoured to 1 rmsd. The backside of PP1G is shown in grey surface presentation. (**D**) Structure of the PP1-binding portion of PPP1R15B from the complex shown in '**B**' aligned to the PP1-binding portions of the regulatory subunits PNUTS (PDB: 4MP0) and spinophilin (PDB: 3EGG). Side-chains of PPP1R15B residues that make important contacts with PP1G and their counterparts in PNUTS and spinophilin are shown in stick diagram.

The following figure supplement is available for figure 2:

**Figure supplement 1**. Variable occupancy of the M1 metal binding site in the PP1G-PPP1R15B binary complexes.

bacterially-expressed PERK (*Marciniak et al., 2006*), served as the specific substrate. Surprisingly, across a range of enzyme concentrations over which the reaction velocity was enzyme-limited (6.5–25 nM), the non-specific $GST^P$ substrate (at 1.7 µM) was more rapidly dephosphorylated by the PPP1R15B-PP1G binary complex than the specific $eIF2a^P$ substrate (despite the latter being present at a higher concentration, 2.8 µM) (*Figure 3B*). A similar hierarchy was observed with reactions in which the concentration of binary complex was fixed (at 26 nM) and the concentration of substrate varied: across a range of substrate concentrations over which the reaction velocity was substrate-limited (0.5–3 µM) the non-specific $GST^P$ substrate was threefold (95% confidence limits 3.6 to 2.6) more rapidly dephosphorylated by the PPP1R15B-PP1G binary complex than the specific $eIF2a^P$ substrate (*Figure 3C,D*). These differences in velocity were observed at early time points of the reaction (*Figure 3D*) and persisted into later time points (with considerable substrate depletion), obeying first order kinetics with consistently more rapid dephosphorylation of $GST^P$ compared with $eIF2a^P$ (*Figure 3E*).

A binary complex produced with PP1G and the homologous region of mouse PPP1R15A (539–614) similarly proved more effective at dephosphorylating the non-specific $GST^P$ substrate (*Figure 3—figure supplement 1*). Phosphatase activity was strictly dependent on the presence of both the regulatory and catalytic subunits in the *E. coli* expression system (*Figure 3—figure supplement 2*), leading to the conclusion that binary complexes of PP1 and the in vivo-defined functional core of PPP1R15 possess no measureable selectivity toward $eIF2a^P$ over $GST^P$ in vitro.

Given that the functional core of PPP1R15 promotes $eIF2a^P$ dephosphorylation in cells, these observations suggested that something might have been missing from the complex constituted of bacterially-expressed proteins. Therefore, we measured the ability of tissue lysates to complement the activity of the binary complex and endow it with substrate-specificity. Mouse pancreas (a tissue in which the ISR plays an important role) was homogenized to obtain a cytoplasmic lysate, which was added in escalating amounts to binary complex (1.5–7.5 nM) and substrate (~2 µM). At the concentrations used (up to 100 ng/µl of protein) the lysate itself had minimal $GST^P$ or $eIF2a^P$-directed phosphatase activity (compare lanes 1 and 4 in *Figure 4A*), however, when added to a binary complex of PPP1R15B and PP1 the lysate selectively increased the $eIF2a^P$-directed phosphatase activity without affecting the dephosphorylation of $GST^P$ (*Figure 4A*). These experiments suggest that the lysate is able to provide ingredient(s) missing from the binary complex that endow it with specificity towards $eIF2a^P$.

A clue to the identity of the missing ingredient was provided by experiments showing that G-actin readily joins PPP1R15 and PP1 to form a ternary complex whose abundance and activity respond to changes in actin dynamics in cells; described in detail in the accompanying manuscript (*Chambers et al., 2015*). In keeping with this clue, we found that cytochalasin D, a low molecular weight natural compound that binds in the cleft between lobes I and III of actin and thus disrupts the interactions of G-actin with many of its binding partners (*Nair et al., 2008*), also reversed the stimulatory activity of lysate on $eIF2a^P$ dephosphorylation by binary complexes (*Figure 4B*). Furthermore, as the cell-based experiments in the accompanying manuscript suggested (*Chambers et al., 2015*), jasplakinolide, a marine toxin that promotes actin polymerization (*Holzinger, 2009*), modestly but consistently

**Table 1.** Structure parameters

| | PP1G-PPP1R15B (631–660) | PP1G-PPP1R15B (631–669) | PP1G-PPP1R15B (631–684) | PP1G-PPP1R15B (631–701)-actin |
|---|---|---|---|---|
| **Data collection** | | | | |
| Synchrotron beamline | Diamond I02 | Diamond I02 | Diamond I03 | Diamond I04-1 |
| Space group | $P2_12_12$ | $P2_12_12$ | $P4_12_12$ | C121 |
| Cell dimensions | | | | |
| a,b,c; Å | 66.8, 67.89, 156.38 | 67.01, 67.86, 156.75 | 67.54, 67.54, 158.01 | 103.9, 149.9, 318.7 |
| $\alpha, \beta, \gamma; °$ | 90, 90, 90 | 90, 90, 90 | 90, 90, 90 | 90, 91.03, 90 |
| Resolution, Å | 51.26-1.61 (1.65-1.61) | 33.94-1.55 (1.58-1.55) | 51.34-1.85 (1.89-1.85) | 82.79-7.88 (8.08-7.88) |
| Rmerge | 0.084 (0.807) | 0.097 (0.737) | 0.094 (0.982) | 0.142 (0.680) |
| Rmeas | 0.101 (0.956) | 0.12 (0.927) | 0.107 (1.118) | 0.199 (0.953) |
| $<I/\sigma (I)>$ | 12.5 (3.0) | 7.4 (1.8) | 9.7 (1.5) | 7.2 (1.6) |
| CC1/2 | 0.997 (0.746) | 0.995 (0.685) | 0.998 (0.914) | 0.980 (0.627) |
| Completeness, % | 99.8 (100) | 92.6 (99.9) | 100 (100) | 98 (99.2) |
| Redundancy | 6.3 (6.7) | 5.1 (4.9) | 7.8 (8.2) | 3.4 (3.5) |
| **Refinement** | | | | |
| Rwork | 0.176 | 0.172 | 0.176 | 0.370 |
| Rfree | 0.203 | 0.203 | 0.222 | 0.400 |
| No. of reflections | 92584 | 96347 | 32078 | 5111 |
| No. of atoms | 5493 | 5701 | 2662 | 28185 |
| Average B-factors | 24.4 | 25.2 | 45.2 | 334 |
| Metal ion occupancies | Chain A: M2 0.95 | Chain A: M2 0.79 M1 0.25 | Chain A: M2 0.76 | n/a |
| | Chain C: M2 0.99 | Chain C: M2 0.90 M1 0.22 | | |
| rms deviations | | | | |
| Bond lengths (Å) | 0.006 | 0.006 | 0.011 | 0.0097 |
| Bond angles (°) | 1.044 | 1.054 | 1.221 | 1.271 |
| Ramachandran favoured region, % | 96.6 | 96.4 | 96.8 | 97.2 |
| Ramachandran outliers, % | 0 | 0 | 0 | 0 |
| MolProbity score(percentile) | 1.23 (98%) | 1.2 (98%) | 1.22 (98%) | 1.58 (100%) |
| PDB code | 4V0V | 4V0W | 4V0X | 4V0U |

antagonized the stimulatory activity of lysate on eIF2a[P] dephosphorylation by binary complexes (*Figure 4C*, compare lanes 5, 7, 9 and 11 with 6, 8, 10 and 12) but had no effect on the non-specific phosphatase activity of the binary complex in the absence of lysate (lanes 1 and 3 vs 2 and 4).

Addition of pure G-actin (whose polymerization was blocked by latrunculin B) selectively stimulated the eIF2a[P]-directed phosphatase activity of the PPP1R15B-PP1 binary complex (*Figure 4D*, lanes 1–6), but had no positive effect on the dephosphorylation of GST[P] (*Figure 4D*, lanes 7–12). The enzymatic activity of the actin-stimulated PPP1R15B-PP1 binary complex was strongly substrate-concentration dependent and could not be saturated with substrate over the concentration range accessible to testing with the available methodology, precluding the extraction of a $K_m$ for the substrate, or a $V_{max}$ (*Figure 4—figure supplement 1A,B*). This is typical of PP1-holophosphatases in that often they cannot be saturated in vitro by their substrate (*MacKintosh, 1993*). The strong linear relationship between the log change of substrate concentration with reaction time indicated that the dephosphorylation of eIF2a[P] by the PPP1R15B-PP1G-actin ternary complex proceeded as a first

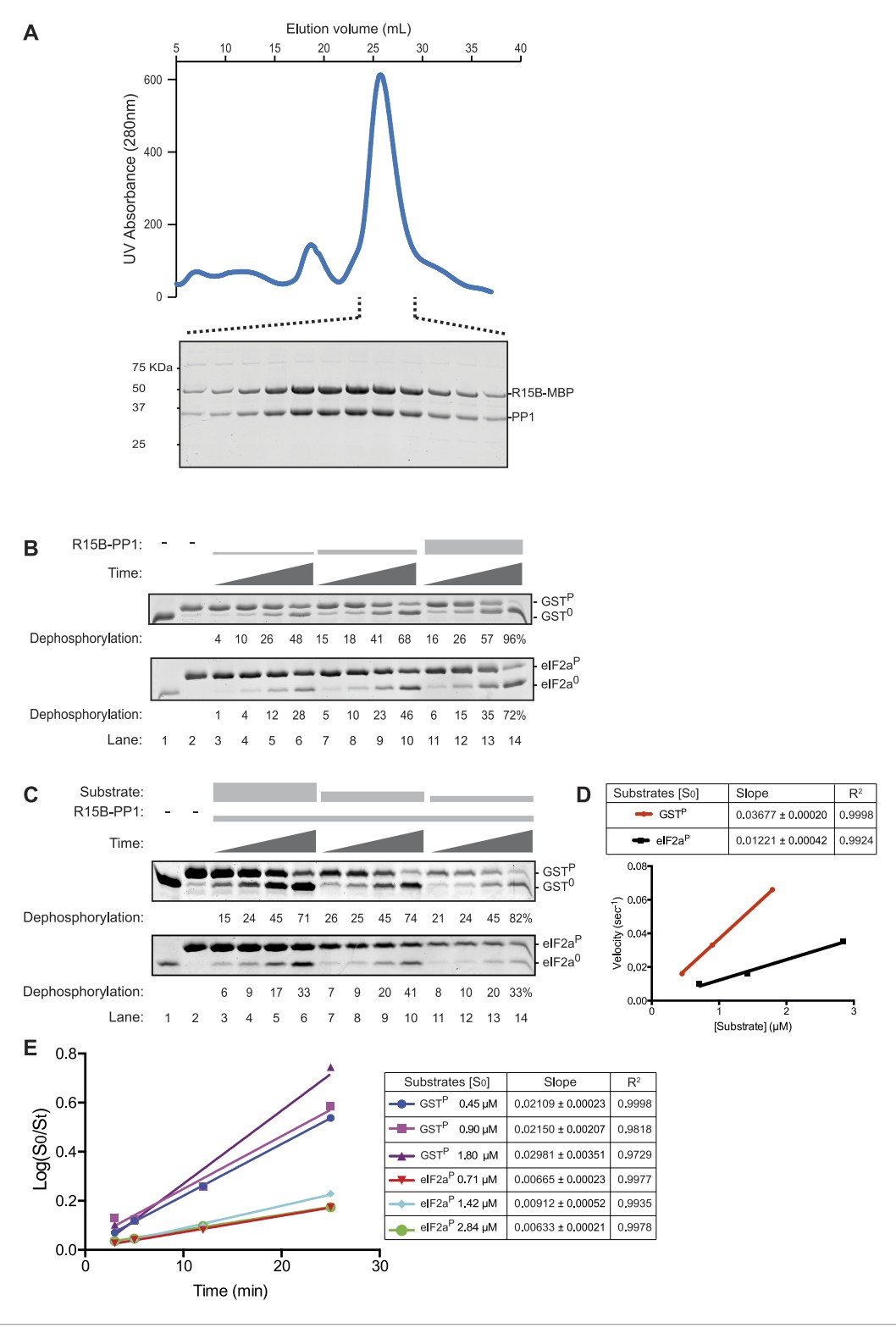

**Figure 3**. The PPP1R15B-PP1 binary complex lacks substrate specificity. (**A**) UV protein absorbance trace of a PPP1R15B-PP1G complex co-expressed in bacteria as a GST-PPP1R15B(631–701)-MBP fusion protein alongside untagged PP1G(7–300), purified by glutathione affinity chromatography and, after cleavage of the GST tag, resolved by size-exclusion chromatography. The indicated fractions from the chromatogram are presented in the Coomassie-stained SDS-PAGE below. The position of PP1G and the PPP1R15B peptide (PPP1R15B-MBP, a fusion with *E. coli*

*Figure 3. continued on next page*

*Figure 3. Continued*

maltose binding protein, MBP, that maintains its stability) are indicated. (**B**) Images of Coomassie-stained Phos-Tag SDS-PAGE in which the non specific substrate and product (GST$^P$ and GST$^0$, upper panel) and the specific substrate and product (eIF2a$^P$ and eIF2a$^0$, lower panel) have been resolved. Escalating concentrations of the binary complex (the enzyme) shown in '**A**' (ranging from ~6.5 nM to 25 nM) have been applied identically to the two substrates (GST$^P$ at 1.7 μM and eIF2a$^P$ at 2.8 μM) for 3, 5, 12, or 25 min. The fraction of the substrate dephosphorylated is indicated under each experimental point. The phosphorylated and unphosphorylated substrates were loaded onto lanes 1 and 2 to serve as a reference for their mobility in the absence of enzyme. Shown is a representative experiment reproduced 4 times. (**C**) As in '**B**'. Dephosphorylation reactions were conducted with a fixed concentration of the binary complex shown in '**A**' [PPP1R15B-MBP and PP1G, at 26 nM] and varying concentration of substrate over a time frame of 3, 5, 12, and 25 min. (**D**) Plot of the initial velocity of the reaction (calculated at the 5 min time point, when excess substrate remains) as a function of substrate concentration derived from the data shown in '**C**'. Note that across a concentration range of substrate over which both reaction velocities are substrate-dependent, the binary complex is more effective at dephosphorylating the non-specific substrate, GST$^P$, than the specific substrate, eIF2a$^P$. Shown is a representative experiment reproduced twice. (**E**) Plot of the logarithm of the time-dependent change in ratio of substrate concentration at t = 0 to the substrate concentration at t [log($S_0/S_t$)] from the six dephosphorylation time courses experiments shown in '**C**'. The slope (mean ± SD), indicative of the relative enzyme velocity, and the linear regression coefficient of the different reactions initiated at the indicated substrate concentration [$S_0$] are noted.

The following figure supplements are available for figure 3:

**Figure supplement 1**. The PPP1R15A-PP1 binary complex also lacks selectivity towards the specific (eIF2a$^P$) substrate over the non-specific (GST$^P$) substrate.

**Figure supplement 2**. Phosphatase activity of the purified bacterially-expressed complexes is dependent on the presence of both the regulatory (PPP1R15A) and catalytic (PP1G) subunits.

---

order process, obeying Michaelis–Menten kinetics for reactions well below the substrate $K_m$ (*Figure 4—figure supplement 1C*). Comparison of the relative velocity of the binary and ternary complex suggested that G-actin stimulated dephosphorylation of eIF2a$^P$ by 13.7-fold (95% confidence limits 11.3–16.9, compare *Figure 3D* with *Figure 4—figure supplement 1B*).

## G-actin, PPP1R15 and PP1 form a ternary holophosphatase that selectively dephosphorylates eIF2a$^P$

To obtain a better physiological perspective on the aforementioned observations pointing to actin's role in stimulating the dephosphorylation of eIF2a$^P$, we measured the substrate concentration in cells by quantitative immunoblotting (using known quantities of purified bacterially-expressed eIF2a to calibrate the assay, *Figure 4—figure supplement 2*). Our estimate of 1.13 ± 0.15 μM eIF2a in the cytosol of HEK 293T cells agrees well with an estimate derived from the number of molecules of the yeast homolog, Sui2p, in a haploid yeast cells (1.71 × 10$^4$ molecules/cell, [*Ghaemmaghami et al., 2003*], which, assuming a yeast cell volume of 60 fl, of which half is cytosol, yields a concentration of 0.95 μM). This information enabled a detailed quantitative comparison of the initial velocity of eIF2a$^P$ dephosphorylation over a range of physiologically relevant substrate concentrations by three different purified enzymes: apo-PP1G, a binary complex of PPP1R15B-PP1G and a ternary complex of actin-PPP1R15B-PP1G. At substrate concentrations in the physiological range (0.5–4 μM), a 16.5-fold acceleration (95% confidence limits 21.2–13.2-fold) of eIF2a$^P$ dephosphorylation was effected by G-actin joining the PP1G-PPP1R15B binary complex to form a ternary complex (*Figure 4E*). Whereas in the absence of actin, PPP1R15B failed to accelerate eIF2a$^P$ dephosphorylation over that observed by apo-PP1 (*Figure 4E*).

In the same assay, actin inhibited the dephosphorylation of a reference substrate, phosphorylase A (PYGM$^P$), by the PPP1R15B-PP1G binary complex 3.1-fold (95% confidence limits 6.8–1.4-fold) (*Figure 4E*); the inhibitory effect of actin on an unstructured substrate, GST$^P$, was less conspicuous (*Figure 4—figure supplement 3*). Together, these observations fit well with prevailing concepts on the basis of substrate-specific dephosphorylation by PP1-holophosphatase complexes, in that their

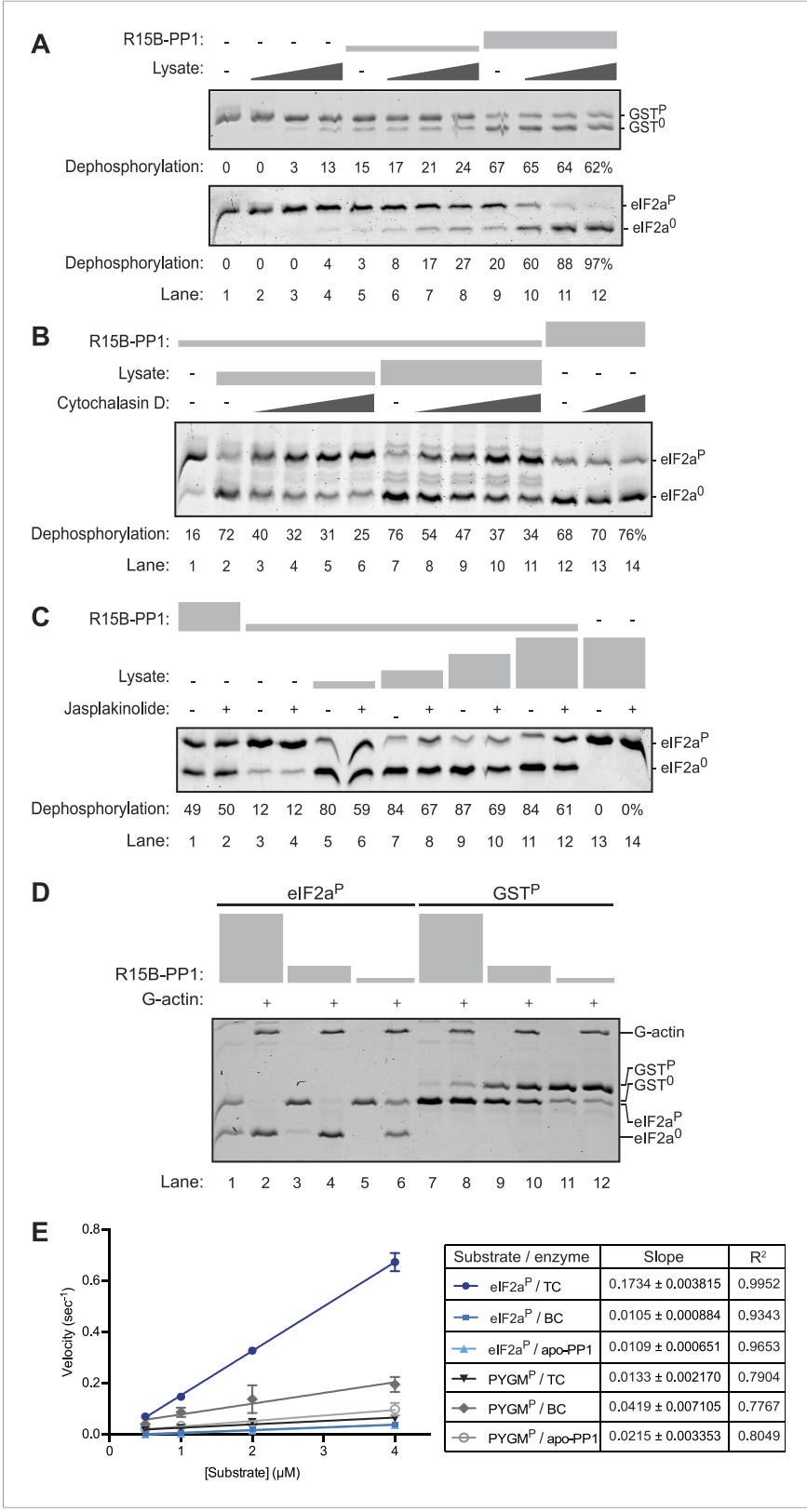

**Figure 4**. An activity in tissue lysate that endows the PPP1R15B-PP1G binary complex with specificity towards eIF2a[P] can be mimicked by pure G-actin. (**A**) Images of Coomassie-stained Phos-Tag SDS-PAGE in which a phosphorylated non-specific substrate and de-phosphorylated product (GST[P] and GST[0], upper panel) and the specific substrate and
*Figure 4. continued on next page*

*Figure 4. Continued*

product (eIF2a$^P$ and eIF2$^0$, lower panel) have been resolved. Escalating concentrations of tissue lysate (ranging from 12–100 ng/µl of protein) were incubated with low (1.5 nM) and high concentration (7.5 nM) of the PPP1R15B-PP1G complex (shown in '*Figure 3A*') and applied identically to the two substrates (at ~2 µM) for 20 min after which the substrate was purified away from other proteins and resolved on the gel shown. The fraction of the substrate dephosphorylated is indicated under each experimental point. (**B**) As in '**A**' with the G-actin-binding compound cytochalasin D added to a final concentration ranging from 12.5 to 100 µM. (**C**) As in '**A**' with the F-actin-stabilizing compound jasplakinolide added at a final concentration of 10 µM. (**D**) As in '**A**' but with pure, latrunculin B-blocked G-actin (~1 µM) in place of tissue lysate. As the entire content of the reaction was loaded (without further purification of the substrate) the actin is visible in this gel, whilst the PPP1R15B-PP1 binary complex, present in quantities below the detection limit of the stained gel, is invisible. (**E**) Plot of the substrate concentration-dependence of the velocity of dephosphorylation of eIF2a$^P$ or phosphorylase A (PYGM$^P$) by apo-PP1G, the binary PP1G-PPP1R15B (BC) or the ternary PP1G-PPP1R15B-actin complex (TC), assayed over a physiologically relevant concentration range of the eIF2a$^P$ substrate (well below the enzyme's $K_m$ for that substrate, see *Figure 4—figure supplement 1*). Shown are the mean ± SD of measurements performed in triplicate. The regression coefficient (R$^2$) and slope (±SD), of the linear regression of each of the six enzyme-substrate pairs are indicated. Note: at substrate concentrations below the $K_m$, the slope of the linear regression reflects the relative velocity of the reaction (proportional to the $K_{cat}/K_m$ of the particular enzyme-substrate pair).

The following figure supplements are available for figure 4:

**Figure supplement 1**. Substrate concentration dependence of the velocity of eIF2a$^P$ dephosphorylation by the PPP1R15B-PP1G-Actin ternary complex.

**Figure supplement 2**. Estimation of the concentration of eIF2a by quantitative immunoblotting of HEK293T cell lysates.

**Figure supplement 3**. Incorporation of G-actin inhibits PPP1R15B-PP1G phosphatase activity directed against non-specific substrates.

---

regulatory component(s), PPP1R15 and G-actin in this case, endow the enzyme with specificity towards its physiological substrate (eIF2a$^P$) whilst inhibiting the dephosphorylation of an irrelevant structured substrate (phosphorylase A) (*Peti et al., 2013*).

Addition of G-actin stimulated the phosphatase activity of both PPP1R15A and PPP1R15B-containing binary complexes. The concentration of G-actin required to elicit a half maximal stimulatory effect was similar for both complexes, <100 nM (*Figure 5A*), whilst an irrelevant protein, bovine serum albumin had no stimulatory effect on eIF2a$^P$ dephosphorylation (*Figure 5—figure supplement 1*). Cytochalasin D and jasplakinolide, which antagonized the stimulatory activity of tissue lysate, also reversed stimulation of the eIF2a$^P$-directed phosphatase activity of the binary complex by purified G-actin (*Figure 5B,C*). The diminishing stimulatory effect of G-actin at the highest concentrations (*Figure 5C*, lanes 9–12) is consistent with high monomer concentrations accelerating F-actin formation during the dephosphorylation reaction (which is conducted at physiological salt concentrations that favor actin polymerization), a feature that is more conspicuous in the jasplakinolide-treated sample.

To further explore the role of G-actin in the holophosphatase complex, we measured the effect of mutations in residues conserved between PPP1R15 proteins on the ability of actin to stimulate eIF2a$^P$-directed phosphatase activity of binary complexes. To compensate for any enfeebling effect the mutations might have on the association of PPP1R15 with PP1 during purification from the bacteria—an important consideration, given the in vivo evidence for cooperativity of actin and PP1 binding (*Chambers et al., 2015*)—we adjusted the concentration of binary complex in the reaction to provide comparable baseline levels of eIF2a$^P$ dephosphorylation (in the absence of added actin). Thus, the concentration of binary complex in the reactions varied from 15–45 nM.

Replacing the conserved residues R$^{658}$ (PPP1R15B) or R$^{571}$ (PPP1R15A), seen to engage the arginine pocket of PP1 with alanine, markedly enfeebled the ability of actin to serve as an activator of substrate specific dephosphorylation (*Figure 6*). Attenuated response to actin was also observed in mutations affecting other conserved residues, W$^{662}$, F$^{672}$, and I$^{676}$ of PPP1R15B (*Figure 6A*) and their counterparts in PPP1R15A (*Figure 6B*). The more severe R$^{571}$A and W$^{575}$A mutations also markedly

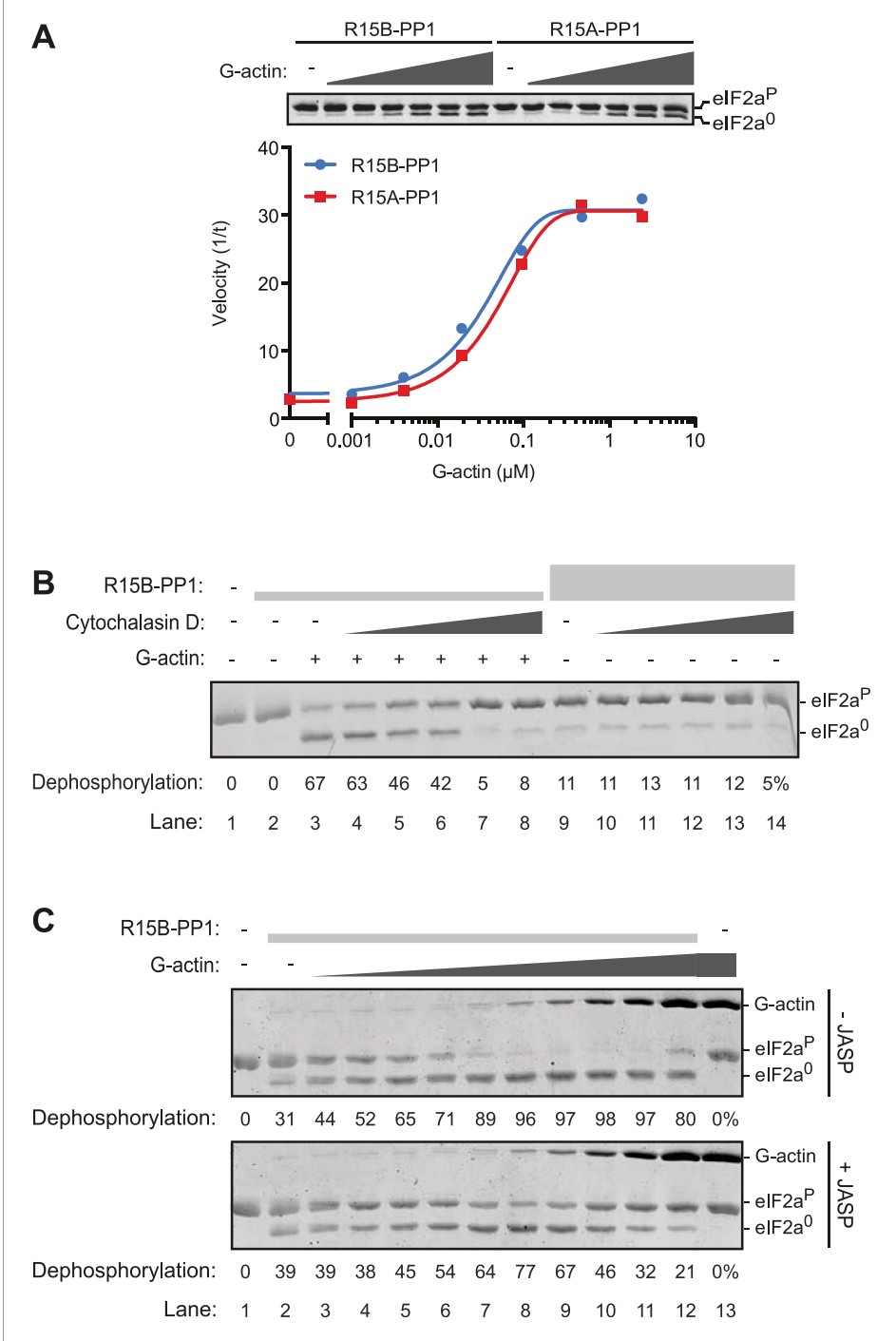

**Figure 5**. G-actin activates the eIF2a$^P$-directed phosphatase activity of both PPP1R15A and PPP1R15B-containing binary complexes with an EC$_{50}$ in the submicromolar range. (**A**) Trace of the velocity of eIF2a$^P$ dephosphorylation by 16 nM PPP1R15A-PP1G (in red) and 68 nM PPP1R15B-PP1G (in blue) binary complexes in the presence of the indicated concentrations of G-actin (fitted to a non-linear regression). An image of the gel of a representative experiment is presented above the trace. (**B**) Image of a Coomassie-stained gel of dephosphorylation reactions by a specific [PPP1R15B-PP1G (25 nM) and actin (1 μM)] ternary complex (lanes 2–8) or a non-specific [PPP1R15B-PP1G (125 nM)] binary complex (lanes 9–14) in the presence of escalating concentrations of

*Figure 5. Continued*

cytochalasin D (1.2–100 µM). (**C**) As in '**B**' but with escalating concentrations of G-actin in the absence and presence of jasplakinolide, 1 µM.

The following figure supplement is available for figure 5:

**Figure supplement 1**. An irrelevant protein, bovine serum albumin, has no effect on the dephosphorylation of eIF2a by the PPP1R15B-PP1G binary complex.

attenuated PPP1R15A's ability to inhibit activation of an ISR target gene in vivo whilst the weaker PPP1R15A F$^{585}$A and I$^{589}$A mutations (that have retained a measure of responsiveness to G-actin in vitro, *Figure 6B*) were indistinguishable from the wildtype in their ability to reverse the ISR (*Figure 6—figure supplement 1*), further supporting the functional significance of the interactions observed in the crystal structures of the binary complex. As the response to actin of the mutant binary complexes was normalized for the recovery of functional catalytic subunit in vitro (reflected in the dephosphorylation activity in the absence of actin), these experiments imply that actin's ability to serve as a substrate-specific activator is dependent on structural features of the ternary complex formed and not merely on anchoring the catalytic subunit to the regulatory one and are in keeping with prevailing concepts on the mechanism of specificity of PP1-containing holophosphatase complexes (*Peti et al., 2013*).

## Structural insights into the G-actin-PPP1R15-PP1 ternary complex and the mode of substrate recruitment

G-actin readily joined binary complexes of GST-PPP1R15 and PP1. Stable ternary complexes were formed by incubating G-actin (purified from rabbit muscle) with bacterially-expressed [GST-PPP1R15 and PP1G] binary complexes immobilized on a glutathione sepharose resin. Cleavage of the GST tag released a complex of the three components in a stoichiometry of 1:1:1, which eluted from a size exclusion column at a position predicted of a trimer (*Figure 7A*, *Figure 7—figure supplement 1*).

Crystals of ternary complexes containing either PPP1R15A or PPP1R15B were obtained, but only the PPP1R15B-containing crystals diffracted X-rays and that to a resolution of only 7.9 Å. Nonetheless, the structure could be solved by molecular replacement, placing five copies of the PPP1R15B-PP1G binary complex (PDB: 4V0U) and G-actin (PDB: 4BIY) (*Mouilleron et al., 2008*) in the crystal unit cell, providing a model of the active holophosphatase (*Figure 7B*, and *Table 1*).

PP1 and actin assemble to form an elongated flat object. The C-terminal structured portion of PPP1R15B faces lobe IV of actin, suggesting that the portion of the regulatory subunit C-terminal of W$^{662}$ that is unstructured in the binary complex is poised to interact with actin and bridge the gap between actin and PP1. Whilst the resolution of the ternary complex is insufficient to trace the regulatory subunit's trajectory beyond that defined by the higher resolution binary complexes (i.e., C-terminal to W$^{662}$), the most significant feature in the averaged difference density map is observed in the cleft between domains I and III in actin's barbed end (*Figure 7C*); suggesting that PPP1R15B extends to engage this site and providing a plausible explanation for the inhibitory effect of cytochalasin D on eIF2a$^P$ dephosphorylation by the ternary complex (*Figure 5B*). The C terminal-most residues of the PPP1R15 functional core (F$^{696}$-Q$^{700}$ in human PPP1R15B and W$^{609}$-R$^{613}$ in mouse PPP1R15A) are good candidates for mediating this interaction. This portion of PPP1R15 proteins can be modeled to form an amphipathic helix, which is found in other G-actin barbed end-binding proteins, exemplified by the drosophila Ciboulot helix (*Dominguez and Holmes, 2011*) (*Figure 7C, D*). And deletion of this portion abrogates actin-mediated acceleration of eIF2a$^P$ dephosphorylation in vitro (*Figure 7E,F*).

Actin lobe II faces away from PP1 and its D-loop is predicted to be free to engage other ligands. Consistent with this prediction, we found that DNase I, which has high affinity for actin's D-loop (*Mannherz et al., 1980*), readily joins actin-PPP1R15-PP1 to form a quaternary complex (*Figure 7—figure supplement 2A*). DNase I binds to the backside of the ternary complex and thus would not be expected to disrupt access to the PP1 active site (*Figure 7—figure supplement 2B*). Consistent with this prediction, we found that a quaternary complex of PPP1R15A-PP1G-Actin-DNaseI retained its ability to de-phosphorylate eIF2a$^P$ (*Figure 7—figure supplement 2C*).

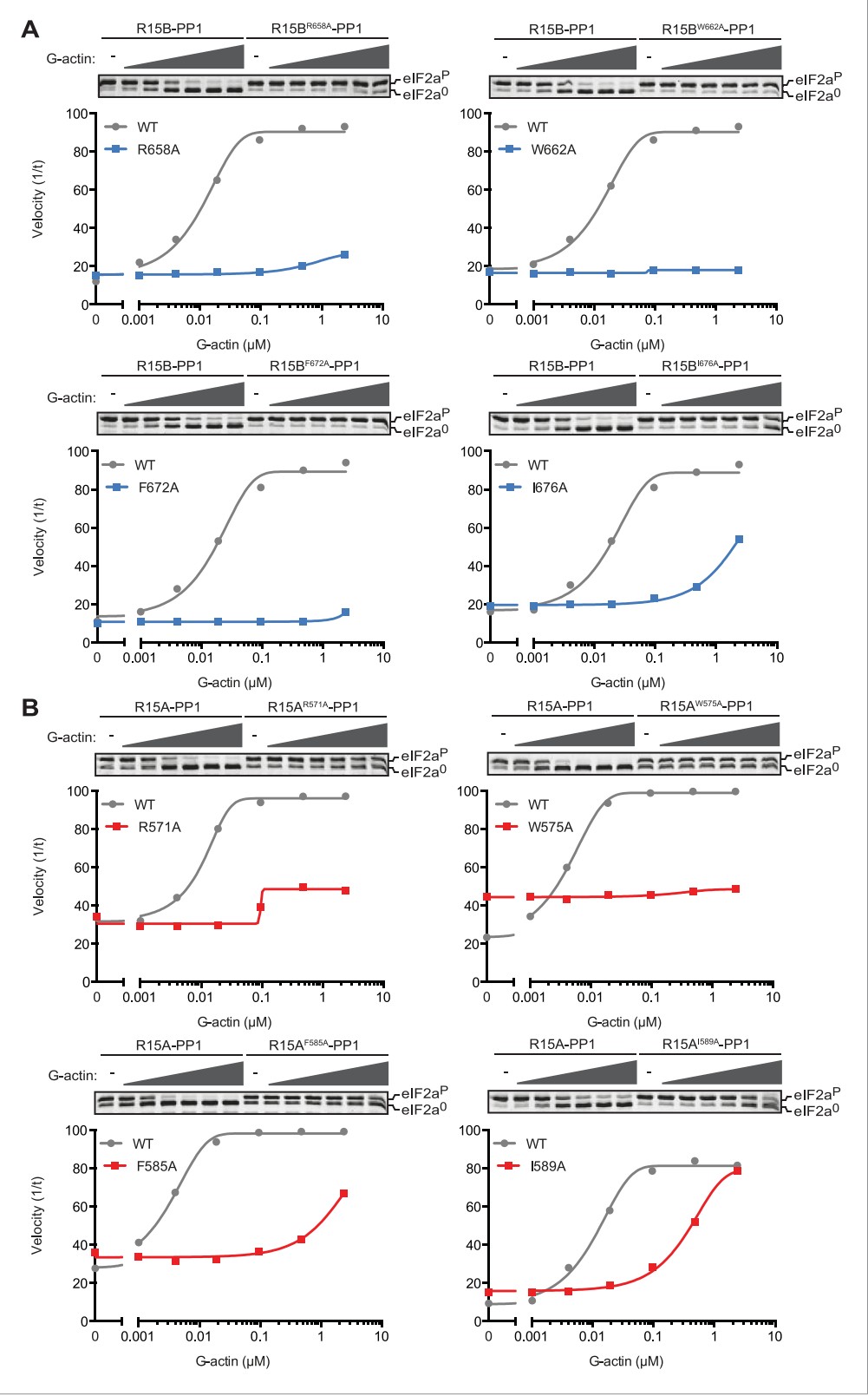

**Figure 6**. Mutations in conserved residues of the PPP1R15 core functional domain enfeeble its activation by actin. (**A**) Traces of the velocity of eIF2a$^P$ dephosphorylation by wildtype and the indicated mutant PPP1R15B-PP1G complexes in the presence of the indicated concentrations of actin. (**B**) As in '**A**' but with wildtype and mutant

*Figure 6. continued on next page*

*Figure 6. Continued*

PPP1R15A-PP1G complexes. Note that the concentration of binary complex varied from 15–45 nM in the assays shown. It was purposely titrated to attain a velocity of dephosphorylation (in the absence of actin) comparable to ~1/5 of that achieved by the wildtype enzyme in the presence of saturating concentration of actin. This ensures comparable activity of the wildtype and mutant enzymes in the absence of actin.

The following figure supplement is available for figure 6:

**Figure supplement 1**. Mutations in conserved residues of PPP1R15A compromise function in vivo.

---

High Ambiguity Driven protein docking with HADDOCK (*Dominguez et al., 2003*) revealed that the platform formed between lobe IV of actin and the catalytic face of PP1 can readily accommodate the N-terminal regulatory lobe of eIF2a (1–185) with phospho-$S^{51}$ inserted deep into the enzyme's active site in proximity to the catalytic metal ions. This mode of binding predicts polar contacts between yeast eIF2a side chains $K^{66}$ and $K^{86}$ ($R^{66}$ and $K^{86}$ in human eIF2a) and PP1 $D^{220}$, eIF2a $E^{92}$ and PP1 $K^{211}$ and a web of hydrogen bonding involving eIF2a $R^{74}$ and $D^{83}$ and actin $D^{222}$ and $N^{225}$ (*Figure 8A*).

To test the importance of these predicted contacts to the dephosphorylation of eIF2a we systematically introduced charge substitutions into the aforementioned residues in eIF2a and compared the ability of the wildtype and mutant proteins to serve as substrates for the non-specific PPP1R15B-PP1G binary complex and actin-containing substrate-specific holophosphatase ternary complex. Mutations $R^{66}E$, $R^{74}E$, $K^{86}E$, and $E^{92}K$ and the $R^{66}E$; $K^{86}E$ double mutant were well expressed and monomeric (*Figure 8—figure supplement 1*) and lent themselves to stoichiometric phosphorylation by active PERK. The $D^{83}K$ mutation compromised solubility and could not be studied further. The $R^{66}E$, $K^{86}E$, and $E^{92}K$ mutations had a modest negative effect on dephosphorylation rates (data not shown). Consistent with contacts made with PP1, the $R^{66}E$; $K^{86}E$ double mutant was compromised ~twofold as a substrate of the binary complex but more than 10-fold as a substrate of the ternary complex (*Figure 8B* and *Figure 8—figure supplement 2*). By contrast the $R^{74}E$ mutation, affecting a predicted contact to G-actin, was selectively compromised in its ability to serve as a substrate of the ternary complex. The extent of this compromise was quantified by a calibrated specificity factor (SF): as the ratio of the dephosphorylation rate of a given substrate at physiological concentration (2 μM) by the ternary complex compared to the binary complex and set to 1 for wildtype eIF2a:

$$SF = \left[ \frac{V_i^{TC(mut)}}{V_i^{BC(mut)}} \right] \div \left[ \frac{V_i^{TC(wt)}}{V_i^{BC(wt)}} \right].$$

The specificity score was lowest for $R^{74}E$, the mutation that was predicted to affect the interaction with actin, whereas the $R^{66}E$; $K^{86}E$ double mutant, which is also compromised in its ability to serve as a substrate of the binary complex, had a higher specificity score (*Figure 8B*, and *Figure 8—figure supplement 2*).

The importance of contacts between eIF2a residues $R^{66}$ and $K^{86}$ and PP1 $D^{220}$ is also supported by the reciprocal $D^{220}K$ mutation in PP1G, which has no measurable effect on the dephosphorylation of the non-specific substrate, $GST^P$, but reproducibly weakens dephosphorylation of $eIF2a^P$ (*Figure 8C, D*). Together, these observations support the plausibility of a binding mode predicted by the computational docking exercise.

## Discussion

Genetics has taught us that the non-redundant, essential, function of the PPP1R15 family is to enable dephosphorylation of eIF2a. It was surprising therefore to learn that the PPP1R15-PP1 complex is as active in dephosphorylating a non-specific substrate as in dephosphorylating $eIF2a^P$. However, a factor found in cell lysates, G-actin, can provide the necessary specificity.

The critical role of G-actin in constituting an $eIF2a^P$-specific holophosphatase is consistent with its emergence as a conserved binding partner of PPP1R15 across phyla and with evidence that manipulations of the cellular cytoskeleton that diminish the pool of free G-actin promote higher levels of phosphorylated eIF2a (*Chambers et al., 2015*). The latter observations point to a role for

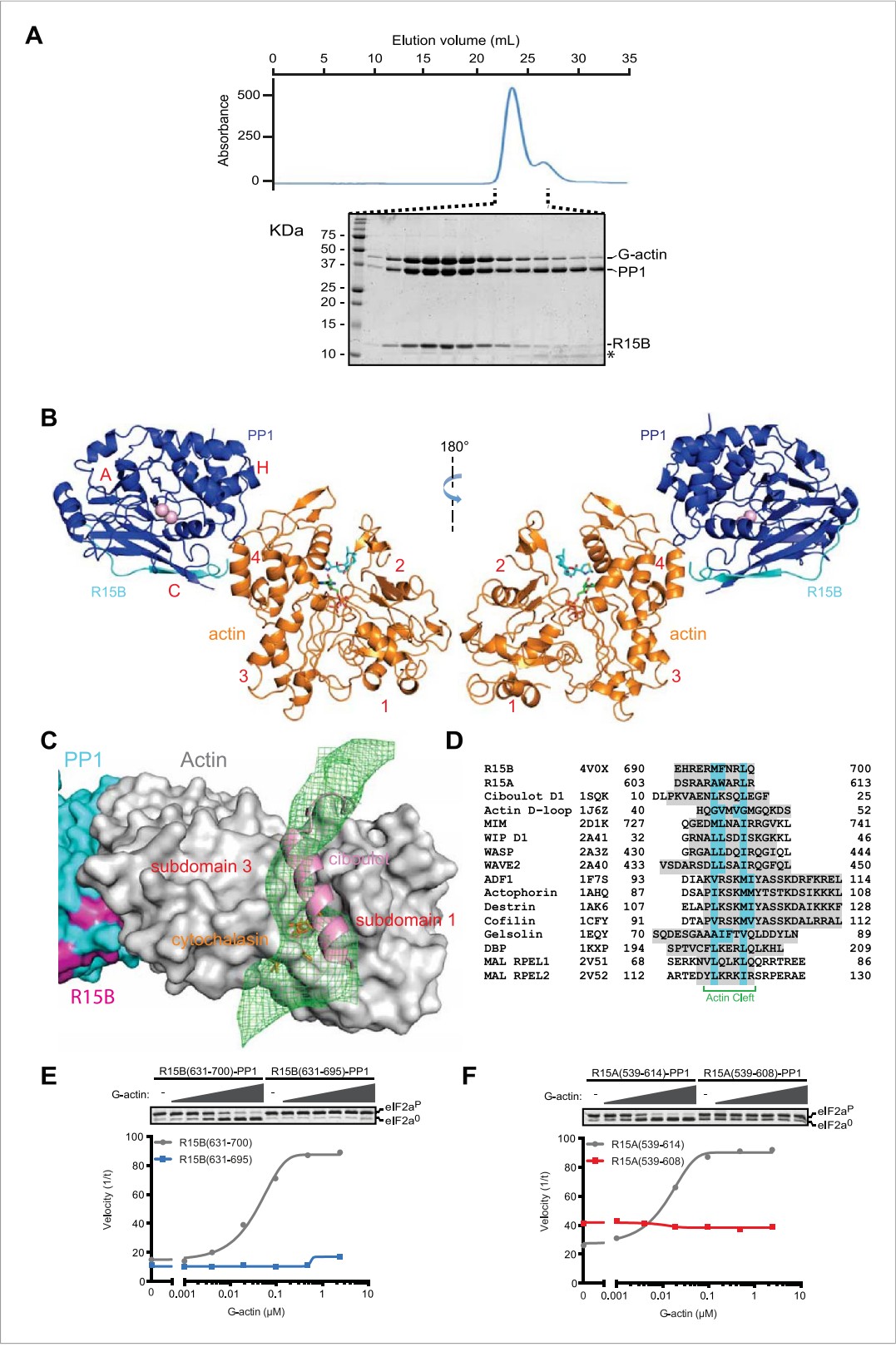

**Figure 7**. G-Actin joins PPP1R15B-PP1G binary complexes to extend its active site-facing surface. (**A**) UV protein absorbance trace of a PPP1R15B(631–701)-PP1G(7–323)-G-actin complex assembled from the bacterially-expressed binary complex and rabbit muscle G-actin and resolved by size-exclusion chromatography. The indicated fractions

*Figure 7. continued on next page*

*Figure 7. Continued*

from the chromatogram are presented in the Coomassie-stained SDS-PAGE below. The position of G-actin, PP1, and the PPP1R15B peptide (R15B) is indicated, as is a degradation product of the PPP1R15B peptide that elutes in later fractions (*). (**B**) Cartoon representation of the PP1G (purple), PPP1R15B (cyan), G-actin (yellow) holophosphatase complex. The peptide backbone is in colored ribbon diagram, the metal ions in the PP1G active site are shown as pink spheres whereas latrunculin B and ATP in the actin nucleotide binding pocket are shown in stick diagram (blue and green respectively). Actin's four lobes (1–4) are marked for reference. The image on the left provides a view of the holoenzyme's active site, whereas the view to the right is of the back side. Note that density attributable to PPP1R15B is only traceable through residue $W^{662}$. (**C**) Close-up of actin's barbed end in the holophosphatase complex. Actin is shown in gray surface representation, PP1 in cyan, and PPP1R15B in magenta. Difference electron density after averaging in coot (likely representing PPP1R15B) is shown as a green mesh. The actin-binding helical peptide of drosophila Ciboulot, a prototypical G-actin binding protein (PDB: 1SQK) (*Hertzog et al., 2004*), in pink ribbon presentation and cytochalasin D, a small molecule inhibitor (PDB: 3EKS) (*Nair et al., 2008*), in orange stick diagram, are provided as landmarks. (**D**) Alignment of the C-terminal most residues of the conserved functional core of human PPP1R15B (R15B) and mouse PPP1R15A (R15A) with amphipathic helices of G-actin binding proteins seen to engage the cleft between domains I and III in the indicated PDB files (shaded grey). The residues observed (or predicted, in the case of PPP1R15) to constitute the hydrophobic face of the amphipathic helix are highlighted in teal. (**E**) Traces of the velocity of $eIF2a^P$ dephosphorylation by wildtype (grey) and the C-terminal truncation mutant PPP1R15B lacking $F^{696}$-$Q^{700}$ (blue) in the presence of the indicated concentrations of actin. (**F**) As in '**D**' but comparing wildtype (grey) and C-terminal truncation mutant PPP1R15A lacking $W^{609}$-$R^{613}$ (red).

The following figure supplements are available for figure 7:

**Figure supplement 1**. G-Actin also joins PPP1R15 A-PP1G binary complexes to form a stable ternary complex.

**Figure supplement 2**. A ternary complex of DNase I, G-actin, PP1G and PPP1R15A retains its $eIF2a^P$-directed phosphatase activity.

cytoskeletal dynamics in regulating rates of $eIF2a^P$ dephosphorylation and are consistent with the finding that most of the stimulatory activity of cell lysate is subject to inhibition by the actin-binding drug cytochalasin D. Nonetheless, our observations are not incompatible with the existence of other cellular proteins contributing to alternative PPP1R15-containing $eIF2a^P$-directed holophosphatases.

Like other PP1 regulators (e.g., spinophilin, inhibitor 2), the functional core of PPP1R15 is natively unstructured (*Yu et al., 2004*) and attains its structure by wrapping around the catalytic subunit. In so doing it follows closely the path of other regulatory subunits, spinophilin (*Ragusa et al., 2010*) and PNUTS (*Choy et al., 2014*). However, these well-resolved interactions (involving residues $R^{639}$-$W^{662}$) account for slightly less than half the length of the PPP1R15B functional core. Residues C-terminal to $W^{662}$, which are not conserved in other regulatory subunits, are also not resolved in crystal structures of the PPP1R15B-PP1G binary complex. Furthermore, binary complexes containing the entire functional core of PPP1R15B require the presence of actin for stability and solubility. Unfortunately, the resolution of the ternary complex is too low to place the portion of PPP1R15B C-terminal of $W^{662}$ in the density maps, but mutagenesis suggests that it plays an important role in actin-mediated substrate-specific dephosphorylation.

The gap between actin and PP1 is potentially suited to accommodate PPP1R15B residues C-terminal to $W^{662}$ whose density is not resolved in the crystal. Furthermore, the presence of density in the cleft between actin domains I and III, which is known to accommodate a short amphipathic helix of other G-actin binding partners (*Dominguez and Holmes, 2011*), is consistent with the engagement of the C-terminal five residues of the functional core of PPP1R15. Deletion of this predicted short amphipathic helical region of PPP1R15 markedly impairs activation by actin in vitro and de-stabilizes actin's association with PPP1R15 in vivo (*Chambers et al., 2015*). Together, the in vivo and in vitro approaches suggest two complementary roles for actin: stabilizing the PPP1R15-PP1 complex and endowing it with substrate specificity. The two functions can be separated in vitro by mutations that retain sufficient PP1 binding to enable reliable measurement of the non-specific phosphatase activity of the binary PPP1R15-PP1 complex and all but eliminate actin's ability to promote substrate specificity.

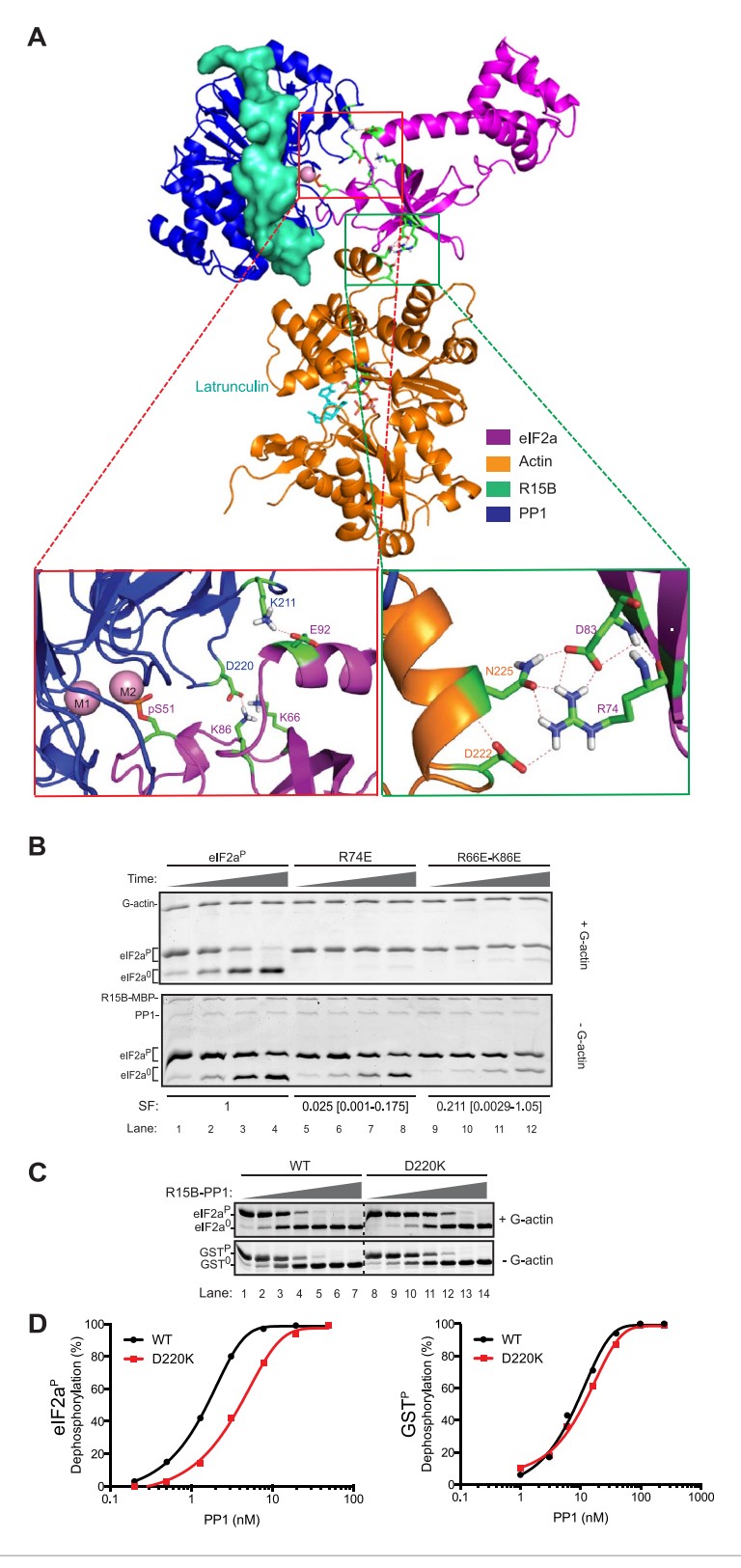

**Figure 8**. Mutation of residues predicted to affect substrate-enzyme binding enfeeble dephosphorylation by the selective ternary complex. (**A**) High ambiguity driven protein docking by HADDOCK (**Dominguez et al., 2003**) model of yeast eIF2a's regulatory N-terminus (PDB: 1Q46) with substrate phospho-S[51] docked at the active site of

*Figure 8. Continued*

the PPP1R15B-PP1-actin ternary complex (PDB: 4V0U). The left close-up view shows the web of predicted hydrogen bonding between eIF2a E$^{92}$ and PP1 K$^{211}$ and eIF2a K$^{66}$ (R$^{66}$ in the mammalian eIF2a) and K$^{86}$ and PP1 D$^{220}$. The right close-up view is of the eIF2a residues R$^{74}$ and D$^{83}$ and actin residues D$^{222}$ and N$^{225}$. (**B**) Images of Coomassie-stained Phos-Tag SDS-PAGE in which phosphorylated and de-phosphorylated wildtype and the indicated eIF2a mutant forms have been resolved. The upper panel ('+G-actin') is of reactions with substrate at ~2 µM exposed to 8.6 nM of the [PPP1R15B-MBP and PP1] binary complex with 1 µM (saturating) concentration of G-actin for 5, 10, 15, and 30 min. In the lower panel ('−G-actin'), the concentration of binary complex was higher (0.26 µM). The specificity factor (SF) of the mutant substrates, based on the relative velocity of dephosphorylation by the PPP1R15B-MBP-PP1-actin ternary complex (TC) compared to the PPP1R15B-MBP-PP1 binary complex (BC) and normalized to the wildtype substrate (see *Figure 8—figure supplement 2*), is noted below each construct (mean and with 95% confidence limits):

$$SF = \left[\frac{V^{TC(mut)}}{V^{BC(mut)}}\right] \div \left[\frac{V^{TC(wt)}}{V^{BC(wt)}}\right].$$

Shown is a typical experiment reproduced four times. (**C**) As above; dephosphorylation reactions performed with escalating concentrations of binary complexes constituted with wildtype or the D$^{220}$K PP1G mutant catalytic subunit and allowed to progress for 30 min. The substrate in the upper panel was wildtype eIF2a$^P$ (~4 µM) and G-actin was added at a saturating concentration. The substrate in the lower panel is the non-specific GST$^P$ (~4 µM). (**D**) Plot of the relationship between dephosphorylation velocity and enzyme concentration for the enzyme-substrate pairs shown in panel '**C**' above.

The following figure supplements are available for figure 8:

**Figure supplement 1**. The wildtype and mutant eIF2a$^P$ substrates have indistinguishable retention profiles on size exclusion chromatography.

**Figure supplement 2**. Kinetic analysis of dephosphorylation reactions shown in *Figure 8B*.

In the ternary complex of PPP1R15-PP1-actin, the active site (of the catalytic subunit) is positioned at the bottom of a shallow platform constituted by actin and PP1. The N-terminal lobe, containing the phosphorylated residue of eIF2a, can be computationally docked into this composite surface, bringing phospho-S$^{51}$ in contact with the catalytic site. This model of the enzyme-bound substrate identifies surface residues in PP1 and actin that could engage the substrate in ionic interactions. In support of this model, we note that mutations in several of these residues enfeeble dephosphorylation of eIF2a$^P$ by the specific ternary complex, but less so by the binary complex. This feature is noted not only for the eIF2a R$^{74}$E mutation, predicted to affect the binding to actin, but also in mutations affecting residues predicted to interact with charged surface residues of PP1 (eIF2a residues R$^{66}$E and K$^{86}$E), suggesting that such substrate-PP1 contacts are especially important in the setting of the actin-containing ternary complex. Together these observations support a simple model whereby the composite surface of actin and PP1, held together by PPP1R15, cooperatively engages its substrate to provide specificity through enhanced affinity of binding.

G-actin reproducibly enfeebled the phosphatase activity of the PP1-PPP1R15 complex towards an irrelevant structured substrate (phosphorylase A). The enfeebling effect of actin is modest by comparison to its stimulatory effect on dephosphorylation of the specific substrate, eIF2a$^P$. This last observation is consistent with the shallowness of the composite surface, which would likely have a modest negative effect on accessibility of a structured substrate to the active site and even less so to inhibit an unstructured phosphorylated peptide (such as GST$^P$). However, as bacterially-expressed PP1 has more promiscuous phosphatase activity than the enzyme purified from natural sources (*Alessi et al., 1993*; *MacKintosh et al., 1996*), we may be underestimating the extent to which G-actin biases against the dephosphorylation of other PP1 substrates, such as phosphorylase A. These considerations may also account for the apparent difference in the extent of the bias against phosphorylase A noted in this study and an earlier one which used PP1 derived from rabbit muscle (*Connor et al., 2001*). The functional importance of an inhibitory facet of regulatory subunit action is influenced by stoichiometric

considerations. As the pool of free PP1 in cells is believed to be low, the dramatic increase in PPP1R15A levels in stressed cells may have consequences not only in terms of enhanced dephosphorylation of eIF2a$^P$ but also in terms of diminished dephosphorylation of other substrates.

Actin is an abundant protein. In vitro, the EC$_{50}$ of G-actin for activation of eIF2a$^P$ dephosphorylation by PP1-PPP1R15 complexes is <100 nM (for both isoforms of PPP1R15). It is therefore possible that under most circumstances G-actin is not limiting to holophosphatase formation. The nearly wildtype ISR-antagonizing activity of the weaker PPP1R15A F$^{585}$A and I$^{589}$A mutants (*Figure 6—figure supplement 1*) observed in the face of a nearly log-order higher EC$_{50}$ for actin activation in vitro (*Figure 6B*), could be explained by high G-actin concentration in CHO cells. However, an informative precedent exists for regulation of cellular processes by variation in the abundance of G-actin: physiological changes in the ratio of monomeric G-actin to filamentous F-actin regulate the activity of serum response factor (SRF) through G-actin's ability to engage SRF's activation partner, MAL/MRTF-A, with affinities in the low micromolar range (*Mouilleron et al., 2008*).

Given the abundance of cellular G-actin-binding proteins, many of which engage the cleft between domains I and III (*Dominguez and Holmes, 2011*) and are thus predicted to compete with PPP1R15, it is possible that in some circumstances G-actin availability may also be limiting to holophosphatase formation and that actin dynamics may thus be coupled to eIF2a$^P$ dephosphorylation. The eIF2a phosphorylation-dependent ISR strongly affects memory formation (*Costa-Mattioli et al., 2005*; *Sidrauski et al., 2013*) and actin dynamics are important in synaptogenesis (*Dillon and Goda, 2005*). Therefore, it is tempting to speculate on localized changes in G-actin levels modulating levels of phosphorylated eIF2 and protein synthesis through localized changes in holophosphatase activity in the dynamic synapse.

The ISR also plays an important role in protein folding homeostasis, especially in the endoplasmic reticulum, where PERK-mediated eIF2a phosphorylation defends the stressed organelle by limiting the influx of newly synthesized proteins. The phenotype of combined deletion of PPP1R15A and PPP1R15B tells us that the capacity to reverse this process, by eIF2a$^P$ dephosphorylation, is essential to homeostasis and cell survival (*Harding et al., 2009*). However, under conditions of severe stress, the normal induction of the *PPP1R15A* gene overshoots its mark, such that genetic (*Marciniak et al., 2004*) or pharmacological (*Boyce et al., 2005*; *Tsaytler et al., 2011*) attenuation of PPP1R15-mediated phosphatase activity is protective. Accordingly, small molecules that engage the composite actin-PP1 surface might serve as specific inhibitors of eIF2a$^P$ dephosphorylation to reverse the aforementioned failure of homeostasis, without affecting the many other PP1-containing holophosphatases in the cell.

## Materials and methods

### Plasmid construction

*Supplementary file 1* lists the plasmids used, their lab names, description and notes their first appearance in the figures and their corresponding label, and provides a published reference, where available.

A combination of PCR-based manipulations, restriction digests, and site-directed mutagenesis procedures was used to mobilize the coding sequence and produces in-frame fusions with the affinity tags (GST, His X6 or FLAG epitope) or mCherry fluorescent tag, and to create the deletions and the point mutations indicated in the text.

### Protein expression and purification

Actin was purified from rabbit muscle as described (*Pardee and Spudich, 1982*), dialyzed against G buffer (2 mM Tris pH 8, 0.2 mM ATP, 0.5 mM DTT, 0.1 mM CaCl$_2$, 1 mM NaN$_3$), blocked from further polymerization by incubation with a fivefold molar excess of Latrunculin B (#428020, Calbiochem), and used in biochemical and structural studies.

Chromatographically purified bovine pancreatic DNase I (>2000 units/mg) was purchased from Worthington Biochemical Corporation (#LS002006, Lakewood NJ) and constituted into a complex with actin as previously described (*Mannherz et al., 1980*).

Binary complexes of mouse PP1G and PPP1R15 with diverse endpoints (*Supplementary file 1*) were created by co-transforming BL21 T7 Express *lysY/I$^q$* E. coli (#C3013, New England Biolabs,

Ipswich, MA) with ampicillin$^r$ marked GST-tagged PPP1R15 expression plasmids and kanamycin$^r$ marked untagged PP1G expression plasmids. Colonies bearing both resistance markers were selected on LB plates with 50 µg/ml kanamycin and 100 µg/ml ampicillin. Both plasmids were stable under this dual selection regime. Cultures of 0.5–6 litres of LB media with 1 mM MnCl$_2$, were inoculated with 1/100 volume of a saturated over-night culture (both under dual selection), allowed to progress to an OD$_{600}$ of 0.6–0.8 at 37°C, at which point they were switched to 18°C and induced with 1 mM IPTG, and cultured further for 20 hr until harvest.

Bacterial pellets were chilled on ice, suspended in 4–8 pellet volumes of ice-cold lysis buffer (50 mM Tris pH 7.4, 500 mM NaCl, 1 mM MnCl$_2$, 1 mM MgCl$_2$, 1 mM TCEP, 100 µM PMSF, 20 mTIU/ml aprotonin, 2 µM leupeptin, and 2 µg/ml pepstatin in 10% glycerol), and lysed with an EmulsiFlex-C3 homogenizer (Avestin, Inc, Ottawa, Ontario) at 4°C. Lysates were clarified in a JA-25.50 rotor (Beckman Coulter) at 33,000×g for 30 min, loaded onto a suspension of glutathione sepharose 4B beads and allowed to bind at 4°C for 1–2 hr. The beads were batch-washed with 45 bed volumes of lysis buffer, transferred to a 10 ml column and further washed with 30 bed volumes of lysis buffer, and eluted in 50 mM Tris pH 7.4, 100 mM NaCl, 40 mM GSH, 0.5 mM MnCl$_2$, 0.5 mM TCEP, 10% glycerol.

Tobacco Etch Virus protease (TEV) cleavage (12.5 µg TEV protease/mg protein) was performed overnight (at 4°C) and the clarified mixture of free GST, PPP1R15-containing complex and residual uncleaved precursor proteins was loaded onto a tandem array of two 10/300 mm columns, Superdex 75 and Superdex 200, with a 1 ml GSTrap 4B column at the outflow (all from GE Healthcare, Buckinghamshire, UK) and developed in gel filtration buffer (20 mM HEPES, 100 mM NaCl, 0.1 mM MgCl$_2$, 0.5 mM MnCl$_2$, 0.1 mM ADP, 0.2 mM TCEP, protease inhibitors). In this configuration, the binary complex elutes first and the free GST and any uncleaved binary complexes are retained in the GSTrap 4B column, eluting later with the glutathione rich 'salt' peak.

Ternary complexes of PP1G-PPP1R15 and actin were assembled by combining stoichiometric amounts of the binary complex (assembled on the glutathione-sepharose resin) with latrunculin B-blocked G-actin, incubated for 90–120 min at 4°C, and eluted with the elution buffer noted above, cleaved and fractionated by the tandem size exclusion chromatography setup described above.

Binary complexes used in biochemical studies only, were purified as described above, with the following modifications: the lysis buffer contained 0.1% Triton X-100 to reduce non-specific binding; the elution buffer contained 20 mM Tris pH 7.4, 100 mM NaCl, 40 mM GSH, 1 mM MnCl$_2$, protease inhibitors, 1 mM TCEP, 10% glycerol; and the TEV cleavage step was omitted in some instances.

PERK kinase domain, N-terminal lobe of eIF2a (with and without an EGFP tag) and GSTag were expressed from plasmids PerkKD-pGEX4T-1, GST_TEV_eIF2a-NM_EGFP, eIF2a-NM_pET30a (and mutant variants), and pGSTag (**Supplementary file 1**) in bacteria, and purified by glutathione sepharose or nickel affinity chromatography accordingly. GST-PERK, assembled on glutathione-sepharose beads was extensively washed in 20 mM Tris pH 7.4, 150 mM NaCl, 4 mM DTT, 0.01% Triton X-100, adjusted to 60% (vol/vol) glycerol. In this configuration, kinase activity is retained for months at −20°C. His X6 tagged proteins were recovered in lysis buffer: 50 mM Tris pH 7.4, 500 mM NaCl, 20 mM imidazole, 1 mM MgCl$_2$, protease inhibitors, 1 mM TCEP, 0.2% Triton X-100 in 10% glycerol, washed in the same and eluted in: 50 mM Tris pH 7.4, 100 mM NaCl, 500 mM imidazole, 1 mM TCEP in 10% glycerol and further purified by size exclusion chromatography (Superdex 75 or 200 10/300, GE Healthcare) in gel filtration buffer: 25 mM Tris pH 7.4, 100 mM NaCl, 0.1 mM EDTA, 1 mM TCEP in 10% glycerol, snap frozen in liquid nitrogen and stored in small aliquots at −80°C.

Mouse pancreas lysate (cytosolic fraction) was produced by homogenization of fresh tissue in a Teflon-glass homogenizer in 4 vol of homogenization buffer (250 mM sucrose, 50 mM Tris–HCl pH 7.4, 5 mM MgCl$_2$, 1 mM DTT, and protease inhibitors) followed by a series of clarification steps: 800×g, 15 min, twice; 6000×g, 15 min, twice, and 100,000×g in a TLA-100 rotor (Beckman Coulter), for 1 hr, all at 4°C. The clarified lysate had a protein concentration of 5–10 mg/ml, based on Bradford's method.

Molar concentrations of solutions of pure proteins were estimated from the UV absorbance spectrum and the extinction coefficient, predicted by the ProtParam tool of ExPasy http://web.expasy.org/protparam/.

## Enzymatic assays

### In vitro phosphorylation of eIF2a and GSTag

PERK and protein kinase A (PKA) were used to phosphorylate eIF2a and GSTag, respectively (**Ron and Dressler, 1992**; **Harding et al., 1999**). Briefly, eIF2a was combined with a suspension of GST-PERK assembled on a glutathione-sepharose resin in the presence of 6 mM MgCl$_2$, 3 mM ATP, 50 mM Tris pH 7.4, 100 mM NaCl, 0.1 mM EDTA, 0.5 mM DTT, and 0.01% Triton X-100 at 37°C for 60 min, resulting in complete conversion of the non-phosphorylated eIF2a$^0$ form to the phosphorylated eIF2a$^P$ form. Phosphorylated eIF2a was then purified by size exclusion chromatography (to eliminate residual PERK kinase and ATP that would obscure the dephosphorylation assay).

### In vitro dephosphorylation of eIF2a$^P$, GST$^P$ and phosphorylase A

Dephosphorylation reactions containing the indicated concentrations of binary complex, G-actin or pancreas lysate, were conducted in assay buffer (50 mM Tris pH 7.4, 100 mM NaCl, 0.1 mM EDTA, 0.01% Triton X-100, 1 mM MgCl$_2$, 1 mM MnCl$_2$, 1 mM DTT in 10% glycerol). Phosphorylated eIF2a$^P$ and GST$^P$, were prepared as described above, phosphorylase A (the active form of the enzyme purified from rabbit muscle in the phosphorylated form on serine 15) was purchased from Sigma (Cat No. P1261). Purified apo-PP1G, binary complexes of PP1G-PPP1R15 and lysate or G-actin were pre-incubated at 30°C for 20 min, shaking (500 rpm) in the absence or presence of Jasplakinolide (Calbiochem #420127), Cytochalasin D (Tocris #1233) or equal volume of vehicle (DMSO). The dephosphorylation reaction was initiated by addition of substrate (eIF2a$^P$, GST$^P$ or PYGM$^P$). The reactions were then incubated at 30°C for the specified time, terminated by boiling the sample in 2% SDS loading buffer, and resolved by 8%, 12.5, or 15% Phos-Tag acrylamide gel electrophoresis (for Phosphorylase A, GST$^P$ and eIF2a$^P$, respectively) according to the manufacturers' instructions (#AAL-107, NARD Institute Ltd., Amagasaki, Japan), stained with Coomassie and imaged on a Odyssey near-infrared imager (LiCor). The intensities of the bands corresponding to the phosphorylated and dephosphorylated species were quantified using ImageJ (NIH) and the ratiometric measurements exploited to calculate substrate and product concentrations.

Enzyme velocity, $V$ was measured at substrate concentrations well below the enzyme's $K_m$ and in samples with less than 25% substrate depletion. Under these conditions, the instantaneous velocity (i.e., rate of substrate conversion to product per molecule of enzyme) is proportional to instantaneous substrate concentration and the equivalent velocity is obtained with the equation below, derived from the integrated rate equation for first order kinetics (**Jencks, 1969**):

$$Vi = \frac{ln\frac{[S]_0}{[S]_f} * [S]_0}{\Delta t * [ENZ]},$$

where, $Vi$ is the initial velocity (the instantaneous velocity at t = 0, with the dimensions of 1/t), $[S]_0$ and $[S]_f$ are, respectively, the substrate concentrations at the beginning and end of the reaction, $\Delta t$ is the time interval of the reaction and $[ENZ]$ is the concentration of enzyme.

As all the assays were performed well below the enzyme's $K_m$ for its substrates, the relationship between $Vi$ and $[S]_0$ was fit to a linear regression whose slope is proportional to the $K_{cat}/K_m$ and thus reflects the relative velocity of the different enzyme-substrate combinations tested.

## Protein crystallization and structural solution

Fractions of PP1G-PPP1R15B binary complex and PP1G-PPP1R15-Actin ternary complex after gel filtration chromatography were pooled for protein concentration to 6–10 mg/ml by centrifugal filter units with molecular weight cutoff of 10 kDa (Millipore, Amico Ultra). Then protein (200 nl) and screen plate reservoir buffer (100 nl) for each drop in 96-well crystallization plate were set up as sitting drops for crystal growth in incubators at either 20°C or 14°C. Crystals of PP1G(7–300)-PPP1R15B(631–660) grew at 20°C in a solution of 2.4 M sodium malonate, pH7.0. Crystals of PP1G (7–300)-PPP1R15B(631–669) grew in 3 M NaCl, 0.1 M Hepes, pH7.5. Crystals of PP1G(7–300)-PPP1R15B(631–684) grew in 2.0 M sodium malonate, pH6.0. Crystals of PP1G(1–323)-PPP1R15B (631–701–H6)-Actin grew over a period of 21 days in solution containing 0.2 M CaCl$_2$, 0.1 M Hepes, pH7.0, 20% PEG6000.

Crystals were harvested with perfluoropolyether cryo oil as cryoprotective agent and diffraction data were collected at 100 K at the Diamond Light Source (see *Table 1* for details of beamlines used).

The data were integrated with either XDS (*Kabsch, 2010*); PP1G(7–300)-PPP1R15B(631–660) and PP1G(1–323)-PPP1R15B(631–701–H6)-actin or iMosflm (*Leslie and Powell, 2007*); PP1G(7–300)-PPP1R15B(631–669) and PP1G(7–300)-PPP1R15B(631–684), then scaled with aimless (*Evans, 2006*) in the CCP4 program suite (*Murshudov et al., 2011*). All structures were solved by molecular replacement in Phaser (*McCoy et al., 2007*). For PP1G(7–300):PPP1R15B(631–660) the structure of PP1G from PDB: 1FJM was used as a model, then after refinement this structure was used as a model for the remaining structures. The actin component of the ternary complex was obtained from PDB: 4B1Y. The three binary complexes were refined using phenix.refine (*Afonine et al., 2012*). The ternary complex was refined with Refmac5 (*Murshudov et al., 2011*); to cope with the low resolution of the data, the refinement used jelly-body and NCS restraints, TLS parameterisation to describe thermal motion, and the actin structure in PDB: 1IJJ was used as an external reference model. Data collection and structure refinement statistics are reported in *Table 1*. Ribbon, surface and density map representations were prepared in the PyMOL Molecular Graphics System (Version 1.5.0.4, Schrödinger, LLC).

## Modeling the substrate-bound quaternary complex

The quaternary complex of the PP1G-PPP1R15B-actin ternary complex with its substrate, the N-terminal lobe of eIF2a was modeled by the HADDOCK server (*Dominguez et al., 2003*). Chains B, F, and G of PDB: 4V0U were used as inputs for the ternary complex and the N-terminal lobe of yeast eIF2a (chain A of PDB: 1Q46) as the input for the substrate. Distance restrains between phosphoserine 51 of yeast eIF2a N-terminal domain and Asn124, His125, Asp208, and the two metal ions in the active site of PP1G from the ternary complex were applied.

## Cell culture, transfection and flow cytometry analysis

Chinese Hamster Ovary (CHO) cells were cultured and electroporated using the Neon transfection system (Life Technologies) as described (*Tsunoda et al., 2014*). The effect of PPP1R15A and PPP1RR15B on the activity of the unfolded protein response was studied by transient transfection of stable *CHOP::GFP* reporter cells (*Novoa et al., 2001*). 12 hr after transfection cells were exposed to 2.5 µg/ml tunicamycin (Calbiochem) for an additional 12 hr to activate *CHOP::GFP* and analyzed by flow cytometry. The data were statistically analyzed using FlowJo. Differences in the median reporter values between the wildtype and mutant variants of PPP1R15A were evaluated by One-way ANOVA, by the medians Kruskal–Wallis test (cells not expressing mCherry and with an intensity above $10^3$ were discarded) using GraphPad Prism version 6.0e (GraphPad Software).

## Imunoblotting of eIF2a in cell lysates

HEK293T cells were resuspended in phosphate buffered saline and pelleted gently (800×*g* at 4˚C). The supernatant was decanted, the cell mass measured by weight and the corresponding packed cell volume derived by the assumption of a buoyant density of 1.005 gm/ml (*Loken and Kubitschek, 1984*). Cells were lysed in homogenization buffer (20 mM HEPES pH = 7.5, 150 mM NaCl, 1% Triton X-100, 1 mM EDTA, 1 mM DTT, 10% glycerol, and protease inhibitors), and the clarified lysate resolved by SDS-PAGE alongside samples with a known mass of recombinant eIF2a (purified from bacteria as described above), immunoblotted with a primary rabbit serum directed to the N-terminus of eIF2a (residues 1–185) (lab name NY1308) and an IRDye fluorescently labeled secondary anti-rabbit IgG (LiCor). The fluorescence signals were detected with an Odyssey near-infrared imager (LiCor) and quantified by ImageJ (NIH).

## Acknowledgements

We are grateful to Isabel Novoa (formerly of the Ron Lab), who first detected the presence of actin in association with PPP1R15A (and left excellent records of her discovery, in 2001), to Purbani Chakrabarti and Giles MW Lewis for technical help with protein purification and crystallization, to other members of our labs at the CIMR for their extensive input and support of the project and to John Kendrick-Jones (from the UK Medical Research Council's Laboratory of Molecular Biology) for advice on G-actin preparation and for his gift of chicken G-actin at the initiation of the project.

We thank the Diamond Light Source for access to beamlines I02, I03, and I04-1 (proposal mx8547) that contributed to the results reported here. Supported by grants from the Wellcome Trust (Wellcome 084812/Z/08/Z and 082961/Z/07/Z) the UK Medical Research Council (G1002610) the European Commission (EU FP7 Beta-Bat No: 277713), a Wellcome Trust Strategic Award for core facilities to the Cambridge Institute for Medical Research (Wellcome 100140). SJM is a Senior Clinical Research Fellow of the UK Medical Research Council, and DR and RJR are Wellcome Trust Principal Research Fellows.

## Additional information

### Competing interests

DR: Reviewing editor, *eLife.* The other authors declare that no competing interests exist.

### Funding

| Funder | Grant reference | Author |
| --- | --- | --- |
| Wellcome Trust | 084812/Z/08/Z | David Ron |
| Medical Research Council (MRC) | G1002610 | Stefan J Marciniak |
| European Commission | EU FP7 Beta-Bat No: 277713 | David Ron |
| Wellcome Trust | 082961/Z/07/Z | Randy J Read |
| Wellcome Trust | Wellcome 100140 | Stefan J Marciniak, Randy J Read, David Ron |

The funders had no role in study design, data collection and interpretation, or the decision to submit the work for publication.

### Author contributions

RC, Carried out expression of an active PPP1R15-PP1 binary complex, purified it, assayed its activity, assembled it in a ternary complex, crystallized the complexes and analyzed the structural data and contributed to the writing and editing of the manuscript, Acquisition of data, Analysis and interpretation of data, Drafting or revising the article; CR, Led the biochemical characterization of the various eIF2a phosphatase complexes and performed all the in-cell experiments, contributed to the writing and editing of the manuscript, Acquisition of data, Analysis and interpretation of data, Drafting or revising the article; YY, Carried out crystal manipulation, diffraction data collection, solution of the structures and analysis of the structural data, and contributed to the writing of the manuscript, Acquisition of data, Analysis and interpretation of data, Drafting or revising the article; AC-C, HJC, HPH, Contributed to design and execution of bacterial expression, purification and biochemical analysis of PPP1R15 and contributed to the editing of the manuscript, Acquisition of data, Analysis and interpretation of data, Drafting or revising the article; SJM, Discovered actin's role in the PPP1R15 complex and suggested that it might lend specificity to the PPP1R15-PP1 binary complex and contributed to the writing of the manuscript, Conception and design, Drafting or revising the article; RJR, Contributed to the strategies for crystallization. Oversaw the solution of the diffraction data and interpretation of the structural information and contributed to the writing of the manuscript, Conception and design, Analysis and interpretation of data, Drafting or revising the article; DR, Conceived and oversaw the study as a whole, wrote the manuscript and designed and constructed expression plasmids for the study, Conception and design, Acquisition of data, Analysis and interpretation of data, Drafting or revising the article, Contributed unpublished essential data or reagents

### Author ORCIDs

Stefan J Marciniak, http://orcid.org/0000-0001-8472-7183
Randy J Read, http://orcid.org/0000-0001-8273-0047
David Ron, http://orcid.org/0000-0002-3014-5636

## Additional files

### Supplementary file

• Supplementary file 1. List of the plasmids used in this study, their unique lab identifier, lab name, description, PMID of the relevant reference (if available), figure in which they first appear and cognate label in figure legend.

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
