## [Decision Letter]

## Editorial note dated 06 November 2014

Thank you for sending your work entitled “G-actin provides substrate-specificity to eukaryotic initiation factor 2α holophosphatases” for consideration at *eLife*. Your article has been evaluated by Vivek Malhotra (Senior editor), a Reviewing editor, and three reviewers, one of whom, Richard Treisman, has agreed to share his identity.

The Reviewing Editor and the reviewers discussed their comments before we reached this decision, and the Reviewing editor has assembled the following comments to help you prepare a revised submission.

All three reviewers, with different degrees of enthusiasm, thought that the work in this manuscript was interesting and potentially important. One of the reviewers, Richard Treisman, had some minor comments, but stated: “This is a super paper. It is of interest as it presents a detailed molecular rationale for the biochemical function of actin in a non-cytoskeletal regulatory complex, adding to a number of recent structures in which G-actin plays a role as a regulator rather than regulate. Publish it!” The other two reviewers were less positive, and they have made a number of substantive criticisms that will need to be addressed directly before we can consider a revised manuscript. We have summarized their criticisms below:

Substantive comments:

Despite the extensive dataset and provocative finding that G-actin might regulate PP1/R15 substrate specificity, the potential impact of the work is undermined (1) by conceptual issues related to basic conclusion that PP1/R15 lacks substrate specificity, and (2) by technical limitations related to the phosphatase assays.

Building on a large body of biochemical studies, and substantiated by X-ray crystal structures of a growing number of PP1 complexes, the substrate specificity of PP1 is thought to occur through its interactions with its many 100s of PP1 binding proteins through both negative and positive selection. Like PP1-MYPT1, PP1-spinophilin and others, the interaction of the PP1 binding protein blocks dephosphorylation of substrates (typically phosphorylase is used), but allows either unaltered dephosphorylation of other substrates (e.g. PP1-spinophilin towards GluR1) or enhanced dephosphorylation of a specific target (e.g. PP1-MYPT toward myosin light chain). In the case of R15 early studies (e.g. [9]) showed that the activity of PP1 towards phosphorylase was blocked by GADD34, but activity towards eIF2α was maintained. Therefore the basic experimental set-up used does not address the selective blockade of non-preferred substrates; this is an important omission (our emphasis). The assessment of PP1 activity is also made more complicated by the known limitation that bacterial PP1 (generated in a Mn^2+^ form) possesses promiscuous activity that is not found for native enzyme. Thus the assay used here measures an activity which is not designed to address the question of what is driving preference for eIF2α (our emphasis). Indeed, an interpretation of the key data in Figure 4 is that addition of the cell lysate blocks the non-specific activity towards phospho-GST, not that it confers any change in eIF2α activity, though much more quantitative kinetic analysis is required to know exactly what is going on. On this point, the other reviewer also wondered why addition of lysate alone did not dephosphorylate the non-specific substrate, when there are many different phosphatases in cell lysates.

Both of the less positive reviewers agreed that there is a general lack of quantification (Vmax and Km data are needed to assess activities) and lack of analysis of replicated data for many of the figures. In short, all the phosphatase assays will need repeating to produce at least 3 replicates. The reviewers recognized that this is not trivial and will take time to set up and complete. Depending on the outcome, the conclusions of the paper and the significance of the structural interactions could change.

1) The phosphatase assays rely exclusively on a gel-shift assay that does not allow for assessment of linear rate kinetics, a requirement for a study of this type.

2) Figure 3 is supposed to show that the bacterial complex of PPP1R15A/PPPIc lacks specificity. But, the last two reactions are not relevant. There is apparently a huge amount of enzyme relative to substrate, and both substrates are completely dephosphorylated. What are the enzyme concentrations used for each reaction? More intermediate concentrations should be shown. Also, 3 time points are shown (5, 15 and 30 minutes) but there seems to be little difference in dephosphorylation between 15 and 30 minutes for either substrate suggesting the enzyme may be dying; the substrate seems to be at a high concentration relative to enzyme (2 micro molar substrate). Time points earlier than 5 minutes should be shown.

3) A high concentration of actin is added to dephoshorylation assays (1 micro molar) and compared to assays with no actin (for example Figure 8). Is this really a specific effect? What happens if 1 micro molar concentration of a different purified protein is added?

4) It is unclear what the 'specificity factor“ calculated in Figure 8 really represents. The concentration of enzyme is different, the substrates are different and it is not clear how a velocity was calculated when some of the reactions have gone to completion at the first time point. For R^66^E the two reactions look identical on the gels shown. Only the R^74^E mutation provides useful information regarding the potential role of the interaction of R15 with G-actin. In contrast to the conclusions made by the authors (see also the Discussion), the other mutations do not seem to have much effect. Indeed there is not much difference between the minus and plus G-actin data in panels B, C and D, which is in contrast to data in earlier figures.

5) Information about the relative amounts of dimeric complexes that were used for the various mutants (in the subsection headed “G-actin, PPP1R15 and PP1 form a ternary holophosphatase that selectively dephosphorylates eIF2α” and Figure 6) should be given.

[Editors' note: further revisions were requested prior to acceptance, as described below.]

## Editorial note dated 19 January 2015

Thank you for resubmitting your work entitled “G-actin provides substrate-specificity to eukaryotic initiation factor 2α holophosphatases” for further consideration at *eLife*. Your revised article has been favorably evaluated by Vivek Malhotra (Senior editor), a member of the Board of Reviewing Editors, and two of the original reviewers. The manuscript has been improved but there are some remaining issues that need to be addressed. One of the reviewers, self-identified as Richard Treisman, stated that the rebuttal was satisfactory. However, the other reviewer who examined the revised manuscript disagreed. He/she, as stated below, felt that the major criticism of the first round that the assays methods were not suitable had not been addressed. We note that although reviewer #3 was unable to re-review the manuscript, he/she also made these same points in his/her critique. As reviewer #2 states below: “The onus would then be on the authors to try to resolve why their assay system deviates from what is accepted for PP1 and related protein phosphatases”. It seems that the only resolution to this is for you to perform additional assays, as indicated in the original critique; without these assays and additional experiment, we will not be able to proceed further with this work for *eLife*. Should you be able to perform these additional experiments, we will need to consult with this reviewer again, but we will do our best to expedite the review.

Reviewer #2 (verbatim):

Previously there were substantive concerns raised about conceptual issues related to how the substrate specificity of PP1/R15 was investigated and also concern that the assay methods were sub-optimal in terms of deriving the needed quantitative kinetic information required for a study of this type. While the authors have made an attempt to address all the reviewers' points in the rebuttal, and have added more data that addresses reproducibility to the revised version of this manuscript, these two main points have not really been adequately addressed. There is no specific rebuttal to substantive comment (1). Rather there is an indirect allusion to this point in the Discussion section and some caveats are added. But the authors seem to miss the point that only through full consideration of previous models for PP1 substrate selection can they set up their assay system to adequately assess the action of the R15 proteins on PP1 activity, and then of the potential effects of G-actin. With respect to substantive comment (2), the addition of the data in Figure 3 that shows a time course with lower total dephosphorylation is a step in the right direction but much of the data still shows very high and non-linear dephosphorylation conditions. Notably, the authors attempt to carry out an initial kinetic analysis and come to the conclusion (subsection headed “Substrate-specificity of the PPP1R15-PP1 complex”) that enzymatic activity could not be saturated with substrate “over the concentration range accessible to testing with the available methodology”. But this was the point of the original criticism, that the assay method was not robust. The authors further suggest that such a situation is not unusual for an enzyme. But they fail to acknowledge that numerous studies of PP1 have obtained Michealis-Menton like kinetics. The onus would then be on the authors to try to resolve why their assay system deviates from what is accepted for PP1 and related protein phosphatases.

Specific comments:

1) Related to the issues of the assay method, in Figure 3 is the amount of the substrate calculated correctly? It looks from the Coomassie stains shown that [GST] is at least as high if not higher in total protein that [eIF2α], but the substrate concentration range in panel D does not agree with this visual inspection.

2) To reiterate the conceptual issues raised in the original review, the authors seem surprised that pGST dephosphorylation is not influenced by R15. While the experimental setup does not compare apo-PP1 for a full comparison, the lack of any effect of R15 is exactly what would be expected for a non-specific substrate.

3) Figure 4, lanes 1-4: this is a confusing experiment and impossible to interpret.

4) Figure 4—figure supplement 1: initial time point was 11% only at the highest substrate level. There are no other linear conditions shown so it is not surprising that kinetic information was not derived.

5) Figure 5: also confusing. There seems to be a biphasic effect of G-actin?

6) Figure 6—figure supplement 1: why is there no effect of F585A and I589A?

7) Figure 7 on: very speculative, considering the low resolution structure.

8) Figure 8: why do eIF2α mutants have some effect only in the presence of G-actin. The proposed substrate binding residues would presumably interact with PP1 irrespective of actin. Again more detailed kinetics are needed to see if there is an effect on substrate affinity.

---

## [Author Response]

## Authors response dated 19 December 2014 to editorial note dated 06 November 2014

Substantive comments:

*[…] Both of the less positive reviewers agreed that there is a general lack of quantification (Vmax and Km data are needed to assess activities) and lack of analysis of replicated data for many of the figures. In short, all the phosphatase assays will need repeating to produce at least 3 replicates. The reviewers recognized that this is not trivial and will take time to set up and complete. Depending on the outcome, the conclusions of the paper and the significance of the structural interactions could change*.

The key issues raised in review are related to the biochemical evidence for positive and negative selection of substrates by the binary complex of PP1-PPP1R15 versus the ternary complex of PP1-PPP1R15-Actin. A more formal quantitative analysis of enzyme kinetics of the different enzyme complexes with regard to eIF2α and non-specific substrates was also deemed important.

The revised version addresses these concerns with new experiments and a detailed quantitative analysis as noted below. The robustness of the measurements has been established through multiple replicates and is further attested to by the smooth fitting of multipoint data in the individual experiments. We have also showcased the reproducibility of our experimental setup by providing the reviewers several examples of duplicates of the experimental data submitted for publication (3B^dup^, 3C^dup^, 3D^dup^, 4S1^dup^ and 8B^dup)^. These are detailed below.

Revised Figure 3 presents a detailed comparison of the dephosphorylation reactions containing escalating concentrations of PP1-PPP1R15 binary complex and either the specific (eIF2α^P^) or the non-specific (GST^P^) substrate. It reveals that over a range of enzyme concentrations that are limiting to the velocity of substrate dephosphorylation, the binary complex more rapidly dephosphorylates the non-specific substrate (GST^P^) compared to the specific one (eIF2α^P^).

Revised Figure 3 presents a detailed comparison of the dephosphorylation reactions containing escalating concentrations of the specific (eIF2α^P^) or the non-specific (GST^P^) substrate and fixed concentration of PP1-PPP1R15 binary complex. It reveals that over a range of substrate concentrations that are limiting to the velocity of dephosphorylation, the binary complex more rapidly dephosphorylates the non-specific substrate (GST^P^) compared to the specific one (eIF2α^P^).

Together, revised Figure 3 prove that that the PP1-PPP1R15 binary complex has no selectivity towards eIF2α^P^ compared to GST^P^.

New Figure 4—figure supplement 1 measures the rate of dephosphorylation of eIF2α^P^ by the ternary complex of PP1 -PPP1R15-Actin across a rangeµM. of substrate concentrations from 0.5 to 15. The velocity of the dephosphorylation reaction is seen to increase linearly with substrate concentration. Thus, over the substrate concentration range accessible to testing, the PP1-PPP1R15-Actin enzyme cannot be saturated. This is not unusual in measurements of enzyme kinetics and whilst it precludes extraction of Km (for substrate) or a Vmax, it does enable us to compare the turnover number of the binary complex of PP1-PPP1R15 and the ternary complex of PP1- PPP1R15-Actin over a range of eIF2α^P^ concentration, proving that the inclusion of G-actin in the enzyme complex selectively accelerates the de-phosphorylation of the specific substrate by a factor of at least 15X.

The reviewers also emphasized the importance of measurements aimed at detecting the reciprocal phenomenon, namely the ability of G-actin to selectively enfeeble the dephosphorylation of non-specific substrates. We have addressed this issue in Figure 9, in which we measured the effect of addition of G-actin on the dephosphorylation of a structured substrate, phosphorylase A (PYGM^P^; phosphorylated on serine 15). In contrast to the marked acceleration of eIF2α^P^ dephosphorylation by the addition of actin, the dephosphorylation of PYGM^P^ was inhibited by G-actin. The extent of inhibition may have been limited by the fact that bacterially-derived PP1 may have intrinsically relaxed substrate specificity, a point emphasized by the knowledgeable reviewers. This has now been incorporated into the Discussion.

Author response image 1.**DOI:**
http://dx.doi.org/10.7554/eLife.04871.029

*1) The phosphatase assays rely exclusively on a gel-shift assay that does not allow for assessment of linear rate kinetics, a requirement for a study of this type*.

The Phos-tag gel-based method we use to distinguish the substrates and products of the dephosphorylation reaction is successfully applied as an endpoint assay and affords an accurate measurement of enzyme kinetics over the linear range. This is revealed by the linear regression coefficients of the correlation between rate of substrate to product conversion (reaction velocity) and substrate concentration (see Figure 3 and Figure 4—figure supplement 1, as examples).

*2)*
Figure 3
*is supposed to show that the bacterial complex of PPP1R15A/PPPIc lacks specificity. But, the last two reactions are not relevant. There is apparently a huge amount of enzyme relative to substrate, and both substrates are completely dephosphorylated. What are the enzyme concentrations used for each reaction? More intermediate concentrations should be shown. Also, 3 time points are shown (5, 15 and 30 minutes) but there seems to be little difference in dephosphorylation between 15 and 30 minutes for either substrate suggesting the enzyme may be dying; the substrate seems to be at a high concentration relative to enzyme (2 micro molar substrate). Time points earlier than 5 minutes should be shown*.

The experiment has been redone in accordance with the reviewers’ suggestions. Lower concentrations of enzyme and earlier time points were included. The observations, presented in new Figure 3 reveal that over a range of enzyme concentrations that are limiting to the velocity of substrate dephosphorylation, the binary complex more rapidly dephosphorylates the non-specific substrate (GST^P^) compared to the specific one (eIF2α^P^).

*3) A high concentration of actin is added to dephoshorylation assays (1 micro molar) and compared to assays with no actin (for example*
Figure 8*)*. *Is this really a specific effect? What happens if 1 micro molar concentration of a different purified protein is added?*

Figure 5 and Figure 6 present a detailed analysis of the effect of actin at concentrations ranging from low nanomolar to micromolar. As the figures and text make clear G-actin activates the phosphatase complex with an EC_50_ in the 10^-7^ M range. This does not strike us an un-physiologically high effective concentration for an abundant cellular constituent like actin.

New Figure 5—figure supplement 1 shows that bovine serum albumin in similar concentrations has no effect on eIf2α^P^ dephosphorylation.

*4) It is unclear what the 'specificity factor” calculated in*
Figure 8
*really represents. The concentration of enzyme is different, the substrates are different and it is not clear how a velocity was calculated when some of the reactions have gone to completion at the first time point. For R*^*66*^*E the two reactions look identical on the gels shown. Only the R*^*74*^*E mutation provides useful information regarding the potential role of the interaction of R15 with G-actin. In contrast to the conclusions made by the authors (see also fifth paragraph of the Discussion), the other mutations do not seem to have much effect. Indeed there is not much difference between the minus and plus G-actin data in panels B, C and D, which is in contrast to data in earlier figures*.

The experiments shown in Figure 8 have been re-done focusing on the most informative mutations and providing a time series over which the reactions initial velocity can be estimated. We have also revised the text to explain, in detail, the derivation of the ratiometric specificity factor used to quantify the impact of the mutations in eIF2α on its ability to serve as a specific substrate for the PP1-PPP1R15-Actin ternary complex.

As the revised figure makes clear, the concentration of enzyme in the upper (+Actin) and lower panel (-Actin) were set differently to ensure that both dephosphorylation reactions would proceed to a detectable level over the time period tested. However, as the same concentration of enzyme and substrate was used in comparing the different eIF2α mutants in each panel, the ratio of the velocity in the different reactions define the specificity factor as a valid measure of the ability of the mutant eIF2α to serve as a substrate of the specific ternary complex.

The specificity factor (SF) of a mutant substrate (mut) is based on the relative initial velocity (V_i_) of its dephosphorylation by the PPP1R15B -MBP-PP1-actin ternary complex (TC) versus the PPP1R15B-MBP -PP1 binary complex (BC) and is normalized for the wildtype substrate (at identical enzyme and substrate concentrations):

SF = [V_i_^TC(mut)^÷Vi^BC(mut)^] ÷ [V_i_^TC(wt)^÷V_i_^BC(wt)^]

*5) Information about the relative amounts of dimeric complexes that were used for the various mutants (in the subsection headed “G-actin, PPP1R15 and PP1 form a ternary holophosphatase that selectively dephosphorylates eIF2α” and*
Figure 6*) should be given*.

This information is now provided in the figure legend and on the Results section.

[Editors' note: further revisions were requested prior to acceptance, as described below.]

## Author response dated 06 March 2015 to editorial note dated 19 January 2015

*The manuscript has been improved but there are some remaining issues that need to be addressed. One of the reviewers, self-identified as Richard Treisman, stated that the rebuttal was satisfactory. However, the other reviewer who examined the revised manuscript disagreed. He/she, as stated below, felt that the major criticism of the first round that the assays methods were not suitable had not been addressed. We note that although reviewer #3 was unable to re-review the manuscript, he/she also made these same points in his/her critique. As reviewer #2 states below: “The onus would then be on the authors to try to resolve why their assay system deviates from what is accepted for PP1 and related protein phosphatases”. It seems that the only resolution to this is for you to perform additional assays, as indicated in the original critique; without these assays and additional experiment, we will not be able to proceed further with this work for* eLife*. Should you be able to perform these additional experiments, we will need to consult with this reviewer again, but we will do our best to expedite the review*.

Reviewer #2 (verbatim):

*Previously there were substantive concerns raised about conceptual issues related to how the substrate specificity of PP1/R15 was investigated and also concern that the assay methods were sub-optimal in terms of deriving the needed quantitative kinetic information required for a study of this type. While the authors have made an attempt to address all the reviewers' points in the rebuttal, and have added more data that addresses reproducibility to the revised version of this manuscript, these two main points have not really been adequately addressed. There is no specific rebuttal to substantive comment (1). Rather there is an indirect allusion to this point in the Discussion section and some caveats are added. But the authors seem to miss the point that only through full consideration of previous models for PP1 substrate selection can they set up their assay system to adequately assess the action of the R15 proteins on PP1 activity, and then of the potential effects of G-actin. With respect to substantive comment (2), the addition of the data in*
Figure 3
*that shows a time course with lower total dephosphorylation is a step in the right direction but much of the data still shows very high and non-linear dephosphorylation conditions. Notably, the authors attempt to carry out an initial kinetic analysis and come to the conclusion (subsection headed “Substrate-specificity of the PPP1R15-PP1 complex”) that enzymatic activity could not be saturated with substrate “over the concentration range accessible to testing with the available methodology”. But this was the point of the original criticism, that the assay method was not robust. The authors further suggest that such a situation is not unusual for an enzyme. But they fail to acknowledge that numerous studies of PP1 have obtained Michealis-Menton like kinetics. The onus would then be on the authors to try to resolve why their assay system deviates from what is accepted for PP1 and related protein phosphatases*.

The revised version submitted herein provides a detailed quantitative measurement of the initial velocity of three different enzymes (apo-PP1,PP1+R15, PP1+R15+G-actin) on the specific substrate, eIF2α^P^, and for comparison information on an irrelevant substrate, phosphorylase A (PYGM^P^)(new Figure 4). These measurements, conducted in triplicate, over a range of physiologically-relevant substrate concentrations (established by measuring the concentration of eIF2α in HEK 293 cells, new Figure 4—figure supplement 2), address all the conceptual points of critique. They establish PP1+R15+G-actin as an eIF2α^P^-specific holophosphatase, with the expected attribute of discrimination against irrelevant substrates and showcase the inability of the R15 regulatory subunit alone (that is in the absence of actin) to accelerate eIF2α^P^ dephosphorylation over what is afforded by apo-PP1. As such, the revised version not only supports the conclusion offered in the title and abstract (that “G-actin provides substrate-specificity to eukaryotic initiation factor 2α holophosphatases”) but also aligns this newly described holophosphatase (the ternary complex of PP1 +R15+G-actin) with the prevailing concepts of substrate- specific dephosphorylation by PP1-containing holoenzymes, thus dealing with the concern, raised in review, that our “system deviates from what is accepted for PP1 and related phosphatases”.

In the new experiments reported on in this revised version, we combine an enzyme (apo-PP1, PP1+R15 or PP1+R15+G-actin) and a substrate (eIF2α^P^ or PYGM^P^) at various concentrations and we measure the rate at which the substrate is converted to product over time by applying samples of the reaction mix at different time points to a PhosTag gel that separates the substrate and product. Time-resolved end point assays such as ours, that can measure the presence of both substrate and product in the reaction, are well suited to measure enzyme kinetics.

In such experiments it is simplest to measure the rate of conversion of substrate to product over the early time points, before substantial substrate depletion has taken place, as later measurements may underestimate enzyme velocity. However, this is less of a problem in reactions conducted well under the substrate *Km*, in which enzyme velocity is related linearly to substrate concentration, as it is a simple matter to correct for the effects of substrate depletion by using the integrated rate equation for first order kinetics to deduce the initial velocity (the instantaneous velocity at t=0) (27). In reactions conducted at substrate concentrations approaching or exceeding the *Km*, in which enzyme velocity approaches its maximum, *V*_*max*_, the non-linear relation of velocity to substrate concentration invalidates such simple corrections. However, as eIF2α^P^ is present at concentrations well below the substrate *Km* of its phosphatase (new Figure 4—figure supplement 1 and Figure 4—figure supplement 2) and as we are conducting the measurements of enzyme velocity in this physiological range of substrate concentration (new Figure 4), the impact of substrate depletion on the accuracy of the measurement of enzyme velocity is insignificant.

To ensure that we are measuring enzyme velocity over physiologically relevant substrate concentrations, we applied quantitative immunoblotting (with a standard curve of recombinant protein) to measure the concentration of eIF2a in the cytosol of mammalian cells and calculated the same for yeast cells (based on information available in a public data base; [16]). The concentration of eIF2α in eukaryotic cells (yeast or cultured HEK293 cells is about 1 µM) (new Figure 4—figure supplement 2). Guided by this data, and in specific response to the critiques of reviewers 2 and 3, we have performed a formal analysis of enzyme velocity over a physiologically-relevant range of substrate concentrations (from 0.5-4 µM), comparing the three enzymes (apo-PP1, PP1+R15, PP1+R15+G-actin). In this assay, conducted in triplicate, the initial velocity of eIF2α^P^ dephosphorylation was increased 16.5 fold (95% confidence limits 21.2-3.2 fold) by the presence of actin in the ternary complex, whereas the dephosphorylation of phosphorylase A was decreased 3.1 fold (95% confidence limits 6.8-1.4 fold) by actin’s presence (new Figure 4).

To place these observations in perspective and thereby take into account the prevailing concepts on holophosphatase action we note (in the subsection headed “G-actin, PPP1R15 and PP1 form a ternary holophosphatase that selectively dephosphorylates eIF2α^P^” of the revised manuscript) that together, these observations fit well with prevailing concepts on the basis of substrate-specific dephosphorylation by PP1-holophosphatase complexes, in that their regulatory component(s), R15 and G-actin in this case, endow the enzyme with specificity towards its physiological substrate (eIF2α^P^) whilst inhibiting the dephosphorylation of an irrelevant substrate (phosphorylase A), citing a recent review from Wolfgang Peti in this regard (46).

Thus we believe the biochemical characterization of the actin-containing and actin-free enzyme provide conclusive experimental evidence that G-actin positively affects the ability of PP1+R15 to dephosphorylate eIF2α^P^, compared to other substrates. In so doing we have also addressed the other point deemed conceptual: We have found, as expected, that when actin joins the holophosphatase complex it inhibits the dephosphorylation of an irrelevant substrate (the selective blockade of non-preferred substrates alluded to in the previous editor’s letter).

In the critique of the previous version, the reviewer writes: “but much of the data still shows very high and non-linear dephosphorylation conditions”. We note that as a measure of prudence the quantitative data used to measure the initial velocity (in Figures 3 and 4, and Figure 4—figure supplement 1) were derived from early time points in which substrate depletion was less than 25%. We also wish to reiterate that reactions conducted well under the substrate *K*_*m*_ proceed as a first order process and thus continue to yield valuable information on their kinetics even as the substrate is depleted (27). This is showcased by the strong linear relationship between the log change of substrate concentration with reaction time, extending into the later time points with considerable substrate depletion (new Figure 3, new Figure 4—figure supplement 1 and new Figure 4—figure supplement 3, new Figure 8—figure supplement 2). These findings indicate that the assays performed here report on enzyme substrate pairs that obey Michaelis-Menten kinetics for reactions well below the substrate *K*_*m*_ and thus serve a valid basis for drawing conclusions regarding their relative velocity.

The reviewer is concerned by our finding that the PP1+R15+G-actin enzyme cannot be saturated with substrate, implying that others have been able to measure saturation of PP1 holophosphatase complexes with substrate (to quote: “But they fail to acknowledge that numerous studies of PP1 have obtained Michealis-Menton like kinetics”).

We believe that our experimental finding that the enzyme (PP1+R15+G-actin) is unsaturated by substrate µM at concentrations of up to 15 (estimated to be >15 fold higher than those present in the cytosol) has no bearing on the validity of the measurements of enzyme velocity or on our conclusion that actin selectively accelerates the dephosphorylation of eIF2α^P^ by PP1+R15. Furthermore, we note that in her paper on the subject, and referring to PP1 holophosphatases, Carol MacKintosh states that: “it is frequently impossible to use protein substrates at high enough concentrations to attain V_max_ conditions, the assays therefore use a fixed subsaturating substrate concentration” (33). Thus it would seem that eIF2α^P^ is not exceptional among PP1 holophosphatase substrates in being present in cells at concentrations below its substrate *K*_*m*_ for the holophosphatase and that PP1+R15+G-actin obeys Michaelis-Menten kinetics for reactions well below the substrate *K*_*m*_ and thus does not raise alarms in regards to our methodology for measuring enzyme kinetics, nor diminish in any way the validity of our conclusions.

*Specific comments*:

*1) Related to the issues of the assay method, in*
Figure 3
*is the amount of the substrate calculated correctly? It looks from the Coomassie stains shown that [GST] is at least as high if not higher in total protein that [eIF2α], but the substrate concentration range in panel D does not agree with this visual inspection*.

Molar concentrations of solutions of pure proteins were estimated from the UV absorbance spectrum and the extinction coefficient, predicted by the ProtParam tool of ExPasy http://web.expasy.org/protparam/ (see end of subsection headed “Protein expression and purification” in the revised manuscript); thus, we are very confident in our measurements of substrate (and enzyme) molarity.

The Phos-tag gels on which substrate and products were resolved were used to extract ratiometric information on the relative concentration of the substrate and product in a given sample. The accuracy of this measurement is not affected by the total mass of protein loaded into each lane, nor by the length of staining of each gel, nor by the intensity by which the different substrates take up the Coomassie dye nor by the fluorescence of the particular protein-dye complex (the property being detected by the LiCor Odyssey scanner). Whilst all of these factors may contribute to differences in intensity of the image produced from the different gels they have no bearing on the quantitation of the conversion of substrate to product. In deference to the reviewer’s concern we refer to this point explicitly in the revised Methods section.

*2) To reiterate the conceptual issues raised in the original review, the authors seem surprised that pGST dephosphorylation is not influenced by R15. While the experimental setup does not compare apo-PP1 for a full comparison, the lack of any effect of R15 is exactly what would be expected for a non-specific substrate*.

The revised version presents a comparison of all three enzymes: apo-PP1, PP1+R15 and PP1+R15+G-actin (new Figure 4; also please see detailed response to conceptual issues above), proving that G-actin is selectively accelerating the dephosphorylation of eIF2α^P^, the subject of this paper.

*3)*
Figure 4*, lanes 1-4: this is a confusing experiment and impossible to interpret*.

We acknowledge (and regret) an error in the layout of Figure 4 (in the previous version) and thank the reviewer for bringing this to our attention. Lanes 2 and 4, designed to measure the effect of jasplakinolide in reactions with no lysate (an essential control), were erroneously labeled as emanating from samples to which lysate had been added. This error has now been rectified and the figure is interpretable in showing that the inhibitory effect of jasplakinolide on eIF2α^P^ dephosphorylation is limited to reactions in which G-actin is present.

*4)*
Figure 4—figure supplement 1*: initial time point was 11% only at the highest substrate level. There are no other linear conditions shown so it is not surprising that kinetic information was not derived*.

As noted in the detailed response to the conceptual concerns, eIF2α^P^ concentration in all the reactions is well below the enzymes’ *K*_*m*_ for this substrate. In this range of substrate concentration, enzyme velocity obeys first order kinetics and the distortive effect of substrate depletion on the estimate of enzyme kinetics can be minimized by using the integrated rate equation for first order kinetics to deduce the initial velocity (27). The enzymatic measurements reported in our paper fall within this regime of Michaelis-Menten kinetics, as reflected by the strong linear relationship between the log change of substrate concentration with reaction time, a linear relationship that extends into the much later time points in which substrate depletion has progressed beyond the (arbitrary) 11% cutoff referred to by the reviewer (new Figure 3, new Figure 4—figure supplement 1, new Figure 4—figure supplement 3, new Figure 8—figure supplement 2). Thus, valuable kinetic information is to be gleaned from all the time points in the assays and not only from their earliest points. Plots of time dependent changes in log[s] are now presented in conjunction with the linear regression coefficient and error estimates. These strongly support the conclusions regarding the selective stimulatory effect of actin on dephosphorylation of eIF2α^P^.

These considerations aside, in new Figure 4, measurements were conducted in triplicate and, as a measure of prudence, the reaction was not allowed to progress beyond 25% substrate depletion at any point. Thus, substrate depletion has not compromised the measurements of enzyme velocity nor the key conclusion regarding the relative velocity of the PP1+R15 and PP1+R15+G-actin complex.

*5)*
Figure 5*: also confusing. There seems to be a biphasic effect of G-actin*.

The diminishing stimulatory effect of G-actin at the highest concentrations (Figure 5, lanes 9-12) is consistent with high actin monomer concentrations accelerating F-actin formation during the dephosphorylation reaction. This feature is more conspicuous in the jasplakinolide-treated sample, where actin filaments are stabilized. Of note, the stock of G-actin was maintained in a low ionic strength buffer, which inhibits polymerization, whereas the dephosphorylation assay is performed at physiological salt concentration that favor the concentration-dependent cooperative process of polymerization that can deplete the monomer. We thank the reviewer for calling our attention to this point, which is now discussed in the subsection headed “G-actin, PPP1R15 and PP1 form a ternary holophosphatase that selectively dephosphorylates eIF2α^P^”.

*6)*
Figure 6—figure supplement 1: *why is there no effect of F585A and I589A?*

As noted in Figure 6, the R15A^F585A^ and R15A^I589A^ mutations retain a measure of responsiveness to G-actin in vitro (their IC_50_ for actin is increased about 10 fold over the wildtype) and are thus weak mutations. The contrasting ability of the strong mutations, R15A^R571A^ and R15A^W575A^ (that have lost all measureable responsiveness to G-actin in vitro) to inhibit the ISR and the inability of the weak mutations to do so (Figure 6—figure supplement 1) fits with differences in mutation strength acting in an in vivo assay that has significant threshold effects: with the weak mutations falling, with the wildtype, on one side of that threshold and the strong mutations falling on the other side. This point is now addressed in the Results section of the revised manuscript (in the subsection headed “G-actin, PPP1R15 and PP1 form a ternary holophosphatase that selectively dephosphorylates eIF2α^P^”) and its potential physiological significance in the Discussion. We thank the reviewer for drawing our attention to this point.

*7)*
Figure 7
*on: very speculative, considering the low resolution structure*.

The methods used in the docking experiment are conventional and are explained in detail and hence their speculative nature lay bare. The future reader is made well aware that the conclusions drawn are no more (but also no less) than a plausible speculation on how this enzyme might work. We believe that plausible speculations are fodder for critical thinking and further experimentation and as such have a place in scientific publications.

*8)*
Figure 8*: why do eIF2α mutants have some effect only in the presence of G-actin. The proposed substrate binding residues would presumably interact with PP1 irrespective of actin. Again more detailed kinetics are needed to see if there is an effect on substrate affinity*.

We have reanalyzed the data in these experiments. As new Figure 8—figure supplement 2 show, the eIF2α^P^ R^66^E- K^86^E compound mutations, affecting predicted contacts between the substrate and PP1 indeed enfeebled its dephosphorylation by the PP1+R15 binary complex, as suggested by reviewer 2. Interestingly it is even more compromised in its dephosphorylation by the PP1 + R15 + actin ternary complex. This observation is consistent with the hypothesized PP1-eIF2α contacts playing an especially important role in the context of coincidental stabilizing interactions between the substrate and actin. Such cooperativity in protein-protein interactions is not uncommon. We thank the reviewer for drawing our attention to these issues, which are now discussed in the revised version (in the subsection headed “Structural insights into the G-actin-PPP1R15-PP1 ternary complex and the mode of substrate recruitment” and in the Discussion).

Whilst we agree that more direct biophysical methods to probe substrate enzyme interactions would enable more comprehensive testing of the model for substrate docking, these lie outside the scope of what is already a very large study.